

# High resolution regional modeling of natural and anthropogenic radiocarbon in the Mediterranean Sea

Mohamed Ayache[1], Jean-Claude Dutay[1], Anne Mouchet[1], Nadine Tisnérat-Laborde[1], Paolo Montagna[2], Toste Tanhua[3], Giuseppe Siani[4], and Philippe Jean-Baptiste[1]

[1]Laboratoire des Sciences du Climat et de l'Environnement LSCE/IPSL, CEA-CNRS-UVSQ, Université Paris-Saclay, 91191 Gif-sur-Yvette, France.
[2]Istituto di Scienze Marine, CNR, Bologna, Italy
[3]GEOMAR Helmholtz Centre for Ocean Research Kiel, Dusternbrooker Weg 20, 24105 Kiel, Germany.
[4]Laboratoire Interactions et Dynamique des Environnements de Surface IDES, UMR 8148 CNRS- Université Paris-Sud 11,Orsay, France.

*Correspondence to:* M Ayache (mohamed.ayache@lsce.ipsl.fr)

**Abstract.** A high-resolution dynamical model (NEMO-MED12) was use to give the first simulation of the distribution of radiocarbon ($^{14}C$) across the whole Mediterranean Sea. The simulation provides a descriptive overview of both the natural pre-bomb $^{14}C$ and the entire anthropogenic radiocarbon transient generated by the atmospheric bomb tests performed in the 1950s and early 1960s. The simulation was run until 2010 to give the post-bomb distribution. The results are compared to available in-situ
measurements and proxy-based reconstructions. The radiocarbon simulation allows an additional and independent test of the dynamical model, NEMO-MED12, and its performance to produce the thermohaline circulation and deep-water ventilation. The model produces a generally realistic distribution of radiocarbon when compared with available in-situ data. The results demonstrate the major influence of the flux of Atlantic water through the strait of Gibraltar on the inter-basin natural radiocarbon distribution, and characterize the ventilation of intermediate and deep water ventilation especially through the propagation
of the anthropogenic radiocarbon signal. We explored the impact of the interannual variability on the radiocarbon distribution during the Eastern Mediterranean transient event (EMT). It reveals a significant increase in $^{14}C$ concentration (by more than 60 ‰) in the Aegean deep water, and at intermediate level (value up to 10 ‰) in the western basin. The model shows that the EMT makes a major contribution to the accumulation of radiocarbon in the eastern Mediterranean deep waters.

## 1 Introduction

The Mediterranean region has been identified as a hot-spot for future climatic changes (Giorgi, 2006; Giorgi and Lionello, 2008; MerMex-Group, 2011; Diffenbaugh and Giorgi, 2012). Because this mid-latitude almost-enclosed sea is surrounded by countries with high population growth to the south and highly industrialized countries to the north, it is under strong anthropogenic pressures. This stress is expected to intensify due to factors such as warming and substantial precipitation decrease (Attané and Courbage, 2004). In the context of global change (IPCC, 2013) we need to improve our understanding of
how changes in the climate and circulation of the Mediterranean Sea interact with the biogeochemical processes that define its functioning.





The Mediterranean Sea can be considered as a "miniature ocean", where global change can be studied at smaller/shorter spatial and temporal scales ($\sim$100 yr compared to more than 1000 yr for the global ocean (e.g., Millot and Taupier-Letage, 2005). The Mediterranean Sea has a well-defined overturning circulation with distinct surface, intermediate and deep water masses circulating in the western and the eastern basins, and varying at interannual time scales. This makes it an excellent test

bed for studying basic processes that will also affect the global thermohaline circulation.

The general circulation of the Mediterranean Sea is driven by the high rates of evaporation (the mean evaporation exceeds the mean precipitation) (Millot and Taupier-Letage, 2005). This process transforms warm Atlantic surface water (AW) into cooler and saltier Mediterranean water (Lascaratos et al., 1999), and leads AW to sink offshore in specific northern zones of the Western and Eastern Basins. The Levantine Intermediate Water (LIW) represents one of the main water masses of the

Mediterranean Sea. It spreads throughout the entire Mediterranean basin at intermediate depths (between $\sim$ 150 and 700 m) (Pinardi and Masetti, 2000), and is the major contributor to the Mediterranean outflow into the North Atlantic (Bryden and Stommel, 1984). Furthermore, the LIW participates in the deep convection processes of the Western Mediterranean deep water (WMDW) occurring in the Gulf of Lions, and the Adriatic sub-basin for the Eastern Mediterranean deep water (EMDW) (Millot and Taupier-Letage, 2005). The formation of deep water in the Mediterranean Sea is also characterized by interannual/decadal

variability such as the Eastern Mediterranean Transient (EMT) event, known to create a major shift in deep water formation in the East Mediterranean Sea (EMed) at the beginning of the 1990s (Roether et al., 1996, 2007; Malanotte-Rizzoli et al., 1999; Lascaratos et al., 1999; Theocharis et al., 1992; Beuvier et al., 2012a). The EMT describes a change in the formation site for EMDW, when it temporarily switched from the Adriatic to the Aegean sub-basin.

In many respects, the most useful diagnostics of the ventilation of the ocean's interior come from geochemical tracers

characterized by simple boundary conditions at the ocean's surface, and conservation in deep water (Key et al., 2004; Sarmiento and Gruber, 2006; Broecker and Peng, 1982). In particular, the passive transient tracers (CFC, $^{14}C$, and tritium) do not affect the water mass densities (as opposed to active tracers such as temperature and salinity). Radiocarbon ($^{14}C$) is an ideal tracer for studying air-sea gas exchange, and for assessing the ventilation rate of the deep water masses at very long timescales (Toggweiler et al., 1989a, b).

Radiocarbon ($^{14}C$) is naturally formed by the reaction between nitrogen atoms in the atmosphere and slow-moving neutrons produced by a whole cascade of nuclear reactions between cosmic radiation and molecules in the upper atmosphere. Radiocarbon is not produced in the ocean's interior. All $^{14}C$ enters the ocean from the atmosphere through gas exchange with surface water with an equilibration time of 7-10 years (Broecker and Peng, 1982; Mahadevan, 2001). Radioactive decay of $^{14}C$ (the half-life being 5730 yr) reduces its concentration over time in the water column. Over the last 150 years, the natural

distribution of radiocarbon has been disturbed by i) the dilution of atmospheric $^{14}C$ by the release of fossil fuel $CO_2$, depleted in $^{14}C$ (the Suess Effect; Suess, 1955), and ii) the production of bomb $^{14}C$ by thermonuclear weapon testing in the late 1950s and early 1960s. The latter strongly increased the $^{14}C$ levels in the atmosphere (Rafter and Fergusson, 1957), and consequently the gradient between surface and subsurface waters (e.g., Broecker et al., 1985).

Knowledge of the timescale of the thermohaline circulation is of central importance in the debate on the sequestration of an-

thropogenic carbon in the deep ocean. The radiocarbon concentration in the oceanic water masses is an invaluable tool allowing



us to study the thermohaline circulation from the seasonal cycle, i.e., the near-surface circulation, vertical transport, and mixing (Naegler, 2009; Muller et al., 2006; Rodgers et al., 1997; Guilderson et al., 1998) at decadal and centennial timescales (e.g., Levin and Hesshaimer, 2000; Stuiver et al., 1983). Radiocarbon plays a crucial role in carbon cycle investigations allowing us to assess the carbon fluxes between reservoirs (e.g., Levin et al., 2010), and the description of the air-sea gas exchange process

(e.g., Wanninkhof, 1992; Sweeney et al., 2007).

Understanding the spatiotemporal variation of radiocarbon (Broecker and Peng, 1982) allows us to determine the ages of different water masses and to establish the overturning timescale and water-mass renewal time for individual basins and the global ocean (e.g., Matsumoto et al., 2004). Unlike other tracers, such as tritium, the $^{14}C$ in ocean surface water is not in equilibrium with atmosphere; this means that the surface ocean does not have the same $^{14}C$ age as the atmosphere (i.e., not

zero age). This difference, also known as the "radiocarbon reservoir age" is caused both by the delay in exchange rates between atmospheric $CO_2$ and the carbonate system (Broecker and Peng, 1982), and the dilution effect due to the mixing of surface waters with intermediate or deep waters depleted in $^{14}C$ during seasonal vertical convection or upwelling, respectively. Indeed, when surface waters are isolated from the atmosphere, the radiocarbon clock begins to tick and $^{14}C$ content of water gradually decays.

Radiocarbon observations have played a crucial role as an experimental tool revealing the spatial and temporal variability of carbon sources and sinks (Roether et al., 1980). Observational programmes (e.g., GEOSECS, WOCE and TTO) have provided snapshots of the large-scale distribution of radiocarbon in the world's oceans. However, few $^{14}C$ measurements have been made in the Mediterranean. Broecker and Gerard (1969) provided the first characterization of the natural radiocarbon in the surface and intermediate waters of the whole Mediterranean Sea from in-situ observations. More studies tried to determine the

sea-surface radiocarbon reservoir ages of the Mediterranean, which are mainly affected by the Atlantic surface waters entering at Gibraltar, and/or by local factors related to freshwater input from rivers (Siani et al., 2000). The first $^{14}C$ reservoir age (360 $\pm$ 80 yr) was calculated by Broecker and Olson (1961) using the pre-bomb shells collected along the Algerian continental shelf. Later, Delibrias (1985) obtained an average $^{14}C$ reservoir age of 350 $\pm$ 35 yr through the analysis of pre-bomb mollusc shells from the French and Algerian shelves. The average marine reservoir age for the whole Mediterranean Sea was estimated

by Siani et al. (2000) to be some 390 $\pm$ 85 yr.

Finally, mollusc shells were also used to yield a more significant dataset for the Mediterranean Sea with a mean sea-surface reservoir $^{14}C$ age of 400 $\pm$ 16 yr (Reimer and McCormac, 2002). More recently, the first annually-resolved sea-surface $^{14}C$ record was obtained from a 50-year-old shallow-water coral (*Cladocora caespitosa*) from the western Mediterranean Sea, covering the pre- and post-bomb period (Tisnérat-Laborde et al., 2013). However, all these observations are discrete both in

time and/or space, they cannot give a clear description of radiocarbon evolution between the past and the actual situation now at either the regional or the global scale.

Although box-models have been extensively used to quantify the radiocarbon inventory (e.g., Broecker and Gerard, 1969; Craig, 1969), their application in deriving the oceanic distribution of radiocarbon is limited due to their very simple parameterization. On the other hand, numerical modelling gives us a clear 4-D description of the water column, which provides an

additional opportunity to better understand the $^{14}C$ distribution in seawater.



Several different ocean models have previously been used to study the global radiocarbon distribution (e.g., Toggweiler et al., 1989a, b; Duffy et al., 1995; Mouchet, 2013). However, these studies used coarse-resolution models which could not represent satisfactorily the critical spatial and temporal scales of circulation in the Mediterranean Sea.

Here, we used a high-resolution regional model (NEMO-MED12, horizontal resolution $1/12°$, $\sim 7$ km) of the entire Mediterranean Sea (Beuvier et al., 2010, 2012a). This model has been used previously for biogeochemical studies (Ayache et al., 2015b; Guyennon et al., 2015; Ayache et al., 2015a; Palmiéri et al., 2015; Ayache et al., 2016) and dynamical application (Soto-Navarro et al., 2014; Beuvier et al., 2012a; Lebeaupin Brossier et al., 2011). Here, we use the model to provide the first simulation of radiocarbon distribution and the related reservoir age. The simulation covers the different states of $^{14}C$ from the steady natural distribution to the Suess effect, the $^{14}C$ bomb peak in the 1960s, and the post-bomb distribution until 2011. Our model results are compared to available $^{14}C$ measurements of seawater and marine carbonates reported by Broecker and Gerard (1969), Stuiver et al. (1983), Siani et al. (2000) and Tanhua et al. (2013), and to a 50-year high resolution $^{14}C$ record obtained from a shallow-water coral specimen (Tisnérat-Laborde et al., 2013).

Our work highlights the impact of anthropogenic perturbation ($^{14}C$ bomb peak and the Suess effect) on the radiocarbon distribution across the whole Mediterranean Sea, as well as the regional response across the different sub-basins. In addition, the simulation provides: i) constraints on the $^{14}C$ air-sea transfer; ii) a descriptive overview of the Mediterranean $^{14}C$ distribution, which gives an additional improvement of in-situ data interpretation, and iii) more perspectives on the impact of the interannual variability of the Mediterranean thermohaline circulation (e.g., EMT event) on the modelled $^{14}C$ distribution.

Furthermore, this study is part of the work under way to assess the robustness of the NEMO-MED12 model, and its use in studying the thermohaline circulation and the biogeochemical cycles in the Mediterranean Sea. The overarching objective of this work is to predict the future evolution of this basin under the increasing anthropogenic pressure.

## 2 Method

### 2.1 Circulation model

The Nucleus for European Modelling of the Ocean (NEMO) is a free surface-ocean circulation model (Madec and NEMO-Team., 2008). Here, it is used in its Mediterranean configuration called NEMO-MED12 (Beuvier et al., 2012a) with a horizontal resolution $1/12°$ ($\sim 7$ km), and 50 vertical z-coordinates ranging from 1 m at the surface to 450 m at depth with partial-step formulation.

NEMO-MED12 covers the whole Mediterranean Sea and includes part of the near Atlantic Ocean (buffer zone) from $11°W$ to $36°E$ and from $30°N$ to $47°N$. The exchange with the Atlantic Ocean occurs through this buffer zone, where 3-D salinity and temperature fields are relaxed to the observed climatology (Beuvier et al., 2012a). The sea surface height (SSH) is restored in the buffer zone from the GLORYS1 reanalysis (Ferry et al., 2010) in order to conserve the Mediterranean Sea water volume. The Black Sea is not explicitly represented in NEMO-MED12 configuration; exchanges with the Black Sea consist of a two-layer flow corresponding to the Dardanelles' net budget estimates of Stanev and Peneva (2002).



The atmospheric forcing of NEMO-MED12 is provided by daily mean fields of momentum, freshwater and heat fluxes from the high resolution atmospheric model (ARPERA) over the period 1958-2013 (Herrmann and Somot, 2008; Herrmann et al., 2010). The sea-surface temperature (SST) and water-flux correction term are applied using ERA-40 (Beuvier et al., 2012a). River runoff is derived from the interannual dataset of Ludwig et al. (2009) and Vörösmarty et al. (1996).

The initial conditions (temperature, salinity) are prescribed from the MedAtlas-II (Rixen et al., 2005; MEDAR-MedAtlas-group, 2002) climatology weighted by a low-pass filter with a time window of 10 years between 1955 and 1965 (Beuvier et al., 2012b). For the buffer zone (west of the Strait of Gibraltar) the initial state is based on the World Ocean Atlas 2005 (Antonov et al., 2006; Locarnini et al., 2006).

This model correctly simulates the main structures of the thermohaline circulation of the Mediterranean Sea, with mecha-

nisms having a realistic timescale compared to observations (Ayache et al., 2015a). In particular, tritium (Ayache et al., 2015a) and helium isotope simulations (Ayache et al., 2015b) have shown that the EMT signal from the Aegean sub-basin is realistically simulated during early 1995. However, some aspects of the model still need to be improved: for instance the too-weak formation of Adriatic Deep Water (AdDW), followed by a low contribution to the EMDW in the Ionian sub-basin. In the western basin, the production of WMDW is reliable, but the spreading of the recently ventilated deep water to the south of the

basin is too weak.

Full details of the model and its parameterizations are reported by Beuvier et al. (2012a, b), Palmiéri et al. (2015) and Ayache et al. (2015a).

## 2.2  The tracer model

The $^{14}C$ distribution in the ocean is often expressed as a delta notation relative to the $^{14}C$/C ratio of the atmosphere ($\triangle^{14}C$ =

($^{14}R$/Rref - 1) × 1000; $^{14}R$ is the $^{14}C$/C ratio of the ocean, and for the purpose of ocean ventilation studies Rref is set to one (Toggweiler et al., 1989a).

Here we use the approach of Toggweiler et al. (1989a, b) in which the ratio $^{14}R$ is transported by the model rather than the individual concentrations of C and $^{14}C$. Several model studies adopted the simplified formulation of Toggweiler et al. (1989a) to describe the transport of $^{14}C$ in the ocean (Mouchet, 2013; Muller et al., 2006; Butzin et al., 2005; Orr et al., 2001; England

and Rahmstorf, 1999; Maier-Reimer et al., 1993).

This approach is based on two main assumptions: i) the Dissolved Inorganic Carbon (DIC) field is constant and homogeneous, and ii) the air-sea fractionation processes and biological activity could be ignored (Mouchet, 2013; Toggweiler et al., 1989a). The first assumption reduces the capacity of the model to estimate the $^{14}C$ inventory and the ocean bomb-$^{14}C$ uptake (Mouchet, 2013) but does not much affect the equilibrium $^{14}C$ distribution in the ocean (Maier-Reimer et al., 1993; Orr et al.,

2001; Mouchet, 2013). Modelled and observed $^{14}C$ may be directly compared since the observed $^{14}R$ ratios are corrected for the isotopic fractionation once converted to the standard $\triangle^{14}C$ notation (Stuiver and Polach, 1977).

This simplified approach is commonly used in model evaluation to critically examine the dynamics of the model (i.e., circulation and ventilation) against in-situ observation; because i) many oceanic $^{14}C$ data were obtained either by measuring



$^{14}C$ in dissolved inorganic carbon in seawater or in corals and mollusc shells, and ii) it can be implemented in the ocean circulation models at relatively low computational cost allowing many sensitivity tests (e.g., Matsumoto et al., 2004).

Radiocarbon is implemented in the model as a passive conservative tracer, which does not affect ocean circulation. Hence its movement can be tracked in an off line mode using the pre-computed transport daily fields (U, V, W) of NEMO-MED12 dynamical model (Beuvier et al., 2012b). A time-step of 20 minutes is applied. The same approach was used to simulate the $\varepsilon$Nd (neodymium) distribution in the Mediterranean Sea (Ayache et al., 2016), and the mantle and crustal helium isotope signature (Ayache et al., 2015b), as well as to model the anthropogenic tritium invasion (Ayache et al., 2015a), and CFC and anthropogenic carbon storage (Palmiéri et al., 2015).

Passive tracers are transported in the Mediterranean using a classical advection-diffusion equation, including the sources and sinks. The equation governing the transport of the dissolved inorganic carbon $^{14}R$ in the ocean is:

$$\frac{\delta}{\delta t} {}^{14}R = -\nabla.(u^{14}R - K.\nabla^{14}R) - \lambda^{14}R, \tag{1}$$

Where $\lambda$ is the radiocarbon decay rate, u the 3-D velocity field, and K the diffusivity tensor. Since radiocarbon is not produced in the ocean, all $^{14}C$ enters the surface water through gas exchange. The radiocarbon flux through the sea-air boundary conditions is proportional to the difference in the ratios between the ocean and the atmosphere (Toggweiler et al., 1989a) and given as:

$$F = \kappa R(^{14}R - {}^{14}Ra), \tag{2}$$

Where $\mathcal{F}$ is the flux out of the ocean, and $^{14}Ra$ is the atmospheric $^{14}C$/C ratio. The transfer velocity $\kappa R$ for the radiocarbon ratio in Eq. (2) is computed as:

$$\kappa R = \frac{\kappa CO_2 K_0}{\overline{C_T}} P^a CO_2 \tag{3}$$

with $\kappa CO_2$ being the carbon dioxide transfer velocity, $K_0$ the solubility of $CO_2$ in seawater taken from Weiss (1974), $P^a CO_2$ the atmospheric $CO_2$ pressure, and $\overline{C_T}$ the average sea-surface dissolved inorganic carbon concentration, classically set to 2 mol $m^{-3}$ (Toggweiler et al., 1989a; Orr et al., 2001; Butzin et al., 2005).

The $CO_2$ transfer velocity is computed with the help of surface-level wind speeds, w (m $s^{-1}$), using the ARPERA forcing (Herrmann and Somot, 2008; Herrmann et al., 2010) following the Wanninkhof (1992) formulation:

$$\kappa CO_2 = k_w \times w^2 \sqrt{660/Sc}, \tag{4}$$

Where Sc is the Schmidt number computed with the model S and T fields.

The value of the empirical coefficient $k_w$ depends on the wind field (Toggweiler et al., 1989a; Wanninkhof, 1992; Naegler,



2009). Sensitivity tests were performed to determine the value of the empirical coefficient $k_w$ among the available values in the literature. Here we have used $k_w = 0.25 \times (0.01/3600)$ s $m^{-1}$ for the radiocarbon simulations in the Mediterranean Sea (wind field from ERA40, (Herrmann et al., 2010)), that value produces the best agreement with available in-situ data for the pre-bomb Mediterranean.

## 2.3 Model initialization and forcing

The natural radiocarbon distribution was first simulated using the atmospheric $^{14}Ra$ = 1; the ocean $^{14}R$ is initially set to a constant value of 0.85 ($\triangle^{14}C$ = -150 ‰, appropriate for the deep ocean; (Key et al., 2004)). An atmospheric $CO_2$ of 280 ppm is prescribed for this steady state simulation. These simulations were integrated for 700 years using a 10-year interval of NEMO-MED12 circulation fields between 1965 and 1974 continuously repeated until they reached a quasi-steady state (i.e., the globally averaged drift was less than 0.001 ‰ per year). This forcing period was selected because it does not include any intense interannual variability, such as the event of the Eastern Mediterranean Transient (EMT, Roether et al. (2007); Schroeder et al. (2008)).

Starting from the end of the pre-industrial equilibrium run, the model was integrated from 1765 to 2011 covering the Suess effect (SUESS, 1955), the entire radiocarbon ($^{14}R$) transient generated by the atmospheric nuclear weapon tests performed in the 1950s and early 1960s as well as the anthropogenic $CO_2$ increase. The $^{14}R$ level in the atmosphere (Fig.1) is taken from Orr et al. (2016) and references cited therein, and the atmospheric $CO_2$ from Orr et al. (2001). The radiocarbon values in the buffer zone are prescribed from a global simulation of radiocarbon by Mouchet et al (2016), using a 3-D profile calculated between 35°N and 55°N and from 0° to 46 °W (sensitivity tests were made to determine this box in the North Atlantic).

We also performed a sensitivity test on the impact of the EMT events on the radiocarbon distribution in the Mediterranean Sea. Two separate simulations were run for the period between 1990 and 2010 (i.e., covering the EMT event that occurred at the beginning of the 1990s). The NoEMT run was performed using the classical atmospheric forcing from ARPERA, as described in Sect. 2.1.

To improve dense-water fluxes through the Cretan Arc during the EMT (1992-1993) the ARPERA forcings were modified over the Aegean sub-basin (Beuvier et al., 2012a), by increasing mean values as done by Herrmann and Somot (2008) for the Gulf of Lions. More specifically, from November to March for the winters 1991-1992 and 1992-1993, daily surface heat loss was increased by 40 W $m^{-2}$, daily water loss by 1.5 mm and the daily wind stress modulus by 0.02 N $m^{-2}$. These changes accelerate the transfer of surface temperature and salinity perturbations into intermediate and deep layers of the Aegean sub-basin, and improve the dense-water formation in the Aegean sub-basin during the EMT, with more intense mixing from winter convection.



## 3  Results

### 3.1  Steady state pre-bomb distribution

The $^{14}C$ model results of the radiocarbon natural distribution for March 1956 are expressed in $\triangle^{14}C$ (Fig.2) and in surface radiocarbon reservoir age (Fig.3). They provide a descriptive overview of the basin-wide distribution of radiocarbon before

the anthropogenic perturbation. Figures 2a, 2b and 2c present the horizontal $^{14}C$ distribution of surface waters (between the surface and 200 m depth), intermediate (between 200 and 600 m) and deep waters (between 600 and 3500 m), respectively. Figures 2d and 2e show the radiocarbon distribution over the whole water column in the Mediterranean along a longitudinal transect for both the eastern and western basins together with in-situ observations from Broecker and Gerard (1969). Figure 3 compares model results of reservoir ages and several marine reservoir $^{14}C$ age data available for the surface water of the

Mediterranean; these data were obtained from pre-bomb calcareous marine shells between 1867 and 1948 and coral *Cladocora caespitosa* (Siani et al., 2000; Reimer and McCormac, 2002; Tisnérat-Laborde et al., 2013).

As illustrated in Figs. 2a and 3, there is a significant geographic heterogeneity in surface water for each sub-basin for "natural" (or pre-bomb) $^{14}C$ obtained both from model results and data. Table 1 shows that overall, the average $\triangle^{14}C$ values are generally lower in the WMed corresponding to older reservoir $^{14}C$ age (402 $\pm$ 27) compared to the EMed (349 $\pm$ 14), the

Adriatic (373 $\pm$ 29) and the Aegean (349 $\pm$ 32) sub-basins that show younger reservoir $^{14}C$ ages than the data of Reimer and McCormac (2002). These figures clearly show the surface inflow of Atlantic waters through the Strait of Gibraltar were progressively enriched during their spreading into the EMed, leading to a relatively higher $\triangle^{14}C$ level in the EMed surface water closer to -46 ‰. For both western and eastern surface water, the model simulates $^{14}C$ concentrations slightly higher than the in-situ observations (Broecker and Gerard, 1969; Siani et al., 2000; Reimer and McCormac, 2002; Tisnérat-Laborde

et al., 2013). A careful comparison between model outputs and seawater observations (1959) reveals a more pronounced disagreement, especially in the EMed surface water where the model overestimates the $\triangle^{14}C$ values by more than 10 ‰ (Fig.4a). However, the lack of more in-situ pre-bomb values greatly limits the comparison between model results and observations.

The model also simulates the rapid decrease of $\triangle^{14}C$ values with depth in the eastern basin, marking a significant vertical gradient and the most negative values of deep-water $\triangle^{14}C$ over the entire Mediterranean Sea (-68 $\pm$ 7 ‰). At depth, the model

simulates low levels of $\triangle^{14}C$ in the eastern basin deep water (average value: -64 ‰ $\pm$ 7.4), significantly lower than those simulated in the WMed deep waters (average value: -48 ‰ $\pm$ 6.9) (Fig. 2c). To conclude, the model reproduced reasonably well the E-W gradient and the mean regional values of radiocarbon age, except for the Aegean sub-basin where the model underestimates the regional mean value (Table 1), but the range in the observations is also high.

### 3.2  Distribution of post-Bomb $^{14}C$

The simulated bomb $^{14}C$ ocean distribution in the whole Mediterranean Sea in March 1977 is illustrated in Fig. 5. The large atmospheric $\triangle^{14}C$ increase is reasonably well captured by the model in the surface layer (values up to 120 ‰) over the whole basin. The lowest values are encountered in the known region of convection and formation of deep and intermediate waters (i.e., Gulf of Lions and Cyprus-Rhodes area; Fig.5a). Figure 5b shows a high concentration of radiocarbon at intermediate



depths mainly in areas with recent water-mass ventilation. The radiocarbon distribution is more uniform in the deep water, except one location where relatively high radiocarbon levels are simulated in the deep layer as a result of mixing with the radiocarbon enriched surface water, particularly in the Cretan Sea (values up to $\pm 70\,‰$) (Fig. 5c).

Figures 5d and 5e show the modelled $\triangle^{14}C$ results along vertical sections in the western and eastern basins compared with in-situ data obtained from seawater samples in the Ionian Sea during the GEOSECS expedition in 1977 (Station 404, 35.24°N, 17.12°E, Stuiver and ostlund, 1983). Similarly to the pre-bomb situation, the $\triangle^{14}C$ values decrease rapidly with depth, exhibiting a significant vertical gradient between the maximum in the surface water of around 120 ‰, and the minimum in the deep water values of around -50 ‰ in the western basin and around -60 ‰ in the eastern basin. The model correctly simulates the $\triangle^{14}C$ vertical distribution in the first 1500 m of the water column, in agreement with observations (Fig. 5e). At depth, the model tends to underestimate the $^{14}C$ penetration in the deep Ionian sub-basin, where it fails to reproduce the high $\triangle^{14}C$ levels associated with EMDW formation (Fig. 4b).

Figure 6 displays the modelled $\triangle^{14}C$ evolution between 1765 and 2008 for surface waters (average depth between 0-10 m in dashed line and between 0-100 m depths in solid line) in the Liguro-Provençal sub-basin, plotted against the in-situ values as reconstructed by Tisnerat-Laborde et al. (2013) from a 50-year old zooxanthellate coral *C. caespitosa* collected alive in 1998 along the coast of Bonassola (44°10'N, 9°36'E, NW Mediterranean, 28 m water depth), and from mollusc shells (Siani et al., 2000), Tisnerat-Laborde, personal communication).

Between 1900 and 1952, the modelled $\triangle^{14}C$ values show a slight decrease of ~12 ‰ resulting from the Suess effect (Druffel and Suess, 1983). The model slightly overestimates the observed pre-bomb mean value (-56 $\pm$ 3 ‰, in 1949-1955 Tisnérat-Laborde et al. (2013)) as noted previously. Between 1952 and 1980, the $\triangle^{14}C$ proxy values increase rapidly from -56 ‰ to almost + 85 ‰ in the Ligurian sub-basin due to a net uptake of atmospheric bomb $^{14}C$.

The model represents well the uptake of bomb $^{14}C$ for the top layer (0-10m) and the sub-surface layer (0-100m) until 1965. Then, a slight difference of $\triangle^{14}C$ is simulated between these two layers, with a higher value in the top layer that is consistent with the observations. These differences are the result of vertical convective mixing (Mahadevan, 2001). The greater is the mixing layer depth, the weaker is the amplitude and the peak is delayed. Afterwards, the $\triangle^{14}C$ values decreased slowly with fluctuations, but reaching a value around +50 ‰ in 2008. This gradual decline of $\triangle^{14}C$ (values up to + 60 ‰) is well simulated in the surface water when we compared the modelled present day (March 2011) distribution of radiocarbon in the surface water (Fig. 7). These results demonstrate that the model simulates the bomb $^{14}C$ uptake in surface and sub-surface water with a realistic timescale comparable to in-situ data, and shows a good consistency between the observed and simulated bomb $\triangle^{14}C$ annual average value (Fig. 6).

Figure 7a shows the modelled present day (March 2011) distribution of radiocarbon in the surface water, against Meteor M84/3 cruise data (Tanhua et al., 2013). The $\triangle^{14}C$ distribution pattern for the surface water is similar to the model outputs obtained for years 1956 and 1977, with the eastern basin generally showing higher $\triangle^{14}C$ values compared to the western basin, except in areas of formation of deep and intermediate waters in the Mediterranean Sea ( the Cyprus-Rhodes area, and in the Gulf of Lions), where the $\triangle^{14}C$ concentration decreases rapidly due to higher vertical convection (Fig. 7).





Figures 7d and 7e, present the simulated radiocarbon content for March 2011, at intermediate and deep depth along a W-E transect together with the available Meteor M84/3 cruise data (Tanhua et al., 2013). The two vertical sections show a $\triangle^{14}C$ maximum in the first 500 m depth ($\triangle^{14}C > 40\,‰$). At deeper depths, $\triangle^{14}C$ values exhibit a significant vertical gradient up to 1500 m (Fig. 8), with low $\triangle^{14}C$ values simulated for the deep waters (values lower than -40‰), except for the central

Levantine (i.e the area south of Crete sub-basin) deep water, where high values $\triangle^{14}C$ are simulated (around -20‰) due to the intense deep convection in this areas.

The model correctly reproduces the $\triangle^{14}C$ content of the surface waters as noted previously, with values similar to observations (values about + 50‰, Fig. 7 and Fig. 8). For the deeper depths, the simulated $\triangle^{14}C$ levels tend to be underestimated by more than 20‰ in the WMed and by about 50‰ in the EMed compared to the observations. This is the result of the

too-weak deep water overflow through the Otranto Strait from the Adriatic into the Ionian sub-basin, and the weak southern penetration of the new WMDW in the simulation compared to the values deduced from in-situ observations (Beuvier et al., 2012a, b; Ayache et al., 2015a). This underestimation leads to excessively low $^{14}C$ average values at depth of the eastern basin. However the model simulates well the $\triangle^{14}C$ values in the surface and deep water of Adriatic sub-bassin (Figure 7a and 7c) compared to Meteor M84/3 cruise data (Tanhua et al., 2013).

## 3.3  The spatial and temporal variability

The temporal variability of radiocarbon distribution was explored as a function of sub-basin location (Fig. 9). Specifically, we compared the annual average $\triangle^{14}C$ time series in different "boxes" following the LIW trajectory from the Levantine sub-basin to the Gulf of Lions (including the Tyrrhenian, Ligurian, Algerian and Cretan sub-basins) for the surface (Fig. 9a), intermediate (Fig. 9b), deep (Fig. 9c) and whole water column (Fig. 9d).

The $\triangle^{14}C$ evolution of the surface water is very similar within the different sub-basins until 1965. Afterwards, Tyrrhenian, Algerian and the Ligurian sub-basins have similar bomb $^{14}C$ peak record, while the Gulf of Lions, Levantine basin and Cretan Sea respond differently to the bomb-signal compared to the other sub-basins. The Levantine/Cretan Sea and the Gulf of Lions show surface values as high as 100‰ and as low as 60‰, respectively (Fig. 9a). The differences between the western and eastern basins are more pronounced at intermediate depths (Fig. 9b), especially between the Cretan Sea and the Gulf of Lions,

which shows an almost 40‰ difference in $\triangle^{14}C$. The Algerian and Ligurian sub-basins are characterized by a very similar $\triangle^{14}C$ evolution through time, showing intermediate values between the Cretan Sea and the Gulf of Lions. The results for the Tyrrhenian sub-basin and Cretan Sea indicate higher transfer in intermediate water compared to other sub-basins. Model outputs for the deep layers (600-3500 m) reveal much higher $\triangle^{14}C$ levels in the Cretan Sea compared to the other locations (Fig. 9c) because it has shallower bottom depth. The $\triangle^{14}C$ difference across the six sub-basins is more pronounced at deeper

depths than at the surface (Fig. 9a) and intermediate layers (Fig. 9b), especially after the $^{14}C$ bomb peak. This difference decreases gradually after 1995, particularly in the surface water where the $\triangle^{14}C$ values are almost the same among the different sub-basins.

The impact of the EMT event on the radiocarbon distribution in the Mediterranean was analyzed by comparing the outputs of two model simulations (shown in Fig. 10 and Fig. 11): "EMT" and "NoEMT" for the years of 1995 and 1999, respectively





(see Sect. 2.3). A substantial penetration of radiocarbon is observed in the deep water south of Crete in 1995, as a consequence of the EMT event that increased bottom $\triangle^{14}C$ valeus by more than 60 ‰, close to $^{14}C$ bomb peak values. On the other hand, the EMT reduces the $\triangle^{14}C$ value in the intermediate waters in the EMed (Fig. 10b). The EMT-related $\triangle^{14}C$ signal in the deep waters decreases gradually after the event, with values around 30 ‰ in 1999 (Fig. 10d). For the WMed (Fig. 10a, 10c), the

contrast is particularly pronounced at intermediate levels, with regional values shifted by almost 10 ‰ between 200 and 800 m depth in the Algerian basin (Fig. 10a), as a consequence of the abrupt change in the eastern basin during the EMT event. As shown in Fig. 11, the shift begins in 1992 in the Levantine sub-basin and reached a 60 ‰ difference in 1995 between these two simulations (Figs. 11a, 11b).

## 4   Discussion

The radiocarbon simulations provide independent and additional constraints on the thermohaline circulation and deep-water ventilation in the Mediterranean Sea. The relatively simple approach of radiocarbon modelling adopted here from Toggweiler et al. (1989a) and Mouchet et al (2016) using a high resolution regional model, led to a realistic simulation of the radiocarbon distribution relative to available in-situ data. It also enables the evaluation of the NEMO-MED12 model performance in the Mediterranean Sea from the seasonal to decadal and centennial timescales. Furthermore, it provides a unique opportunity to

better constrain the variability of the uptake of bomb $^{14}C$ in the whole Mediterranean Sea and to study the impact of important hydrological events such as the Eastern Mediterranean Transient (EMT).

     The modelled radiocarbon distribution is very sensitive to the value of the empirical coefficient ($K_W$) (i.e. is the constant regulating air-sea flux). In this study we have used $K_W = 0.25 \times (0.01/3600)$ s $m^{-1}$, this value led to a better simulation of $\triangle^{14}C$ in the Mediterranean compared to the other estimates available in the literature, i.e., $0.426 \times (0.01/3600)$ s $m^{-1}$

used in global scale simulation (Mouchet et al, 2016; Naegler, 2009). The $K_W$ value depends on the wind field and the upper ocean mixing rate field (Wanninkhof, 1992; Toggweiler et al., 1989a). For the present work we used the wind fields from the ARPERA forcing (Herrmann and Somot, 2008; Herrmann et al., 2010) and the atmospheric $CO_2$ values from Orr et al. (2016). These boundary conditions enabled the model to produce satisfactory simulations of the bomb $^{14}C$ chronology. In particular, the timing of the $\triangle^{14}C$ peak in the surface is consistent with the estimated $^{14}CO_2$ time transfer from the atmosphere to the

ocean in the surface waters (∼10 yr, (Broecker and Peng, 1982) as shown in Fig. 6.

     Unlike the global ocean, where input/output of radiocarbon comes only from the exchange with the atmosphere, in the Mediterranean Sea there is also lateral exchange of $^{14}C$ through the Strait of Gibraltar. Unfortunately, there is no time series data of $^{14}C$ concentration in that area. Hence simulated $^{14}C$ levels in the model's Atlantic water (AW) are determined by damping to global model estimates from Mouchet et al (2016) at the western boundary of the model domain, using the 3-D

profile calculated between 35°N and 55°N and from 0° to 46 °W (sensitivity tests were performed to determine this box). This large box in the North Atlantic gave the most representative signature of radiocarbon during the bomb peak (value up to 140 ‰ in 1980) from the global simulation of Mouchet et al (2016).





The comparison between the model outputs and the $\triangle^{14}C$ values from in-situ data reported by Broecker and Gerard (1969), Stuiver et al. (1983) and Tanhua et al. (2013) reveals a good model performance in simulating the bomb/post-bomb radiocarbon distribution (Fig. 4b, Fig. 8). However the representation of the pre-bomb distribution is more contrasted in the simulation (Fig. 4a). Several issues complicate the simulation of the natural steady state distribution of $^{14}C$ using ocean-model circulation:

i) the uncertainty associated with the radiocarbon surface boundary conditions applied in ocean model experiments, ii) the climatological field to represent the wind forcing, often based on atmospheric model outputs and/or historical data, and iii) the significant changes due to the human activity which affects the radiocarbon distribution in the atmosphere and the ocean (e.g., Suess effect). In addition, the limited spatial and temporal resolution of seawater and carbonate organism measurements during the pre-bomb period limits our understanding of the natural radiocarbon distribution in the Mediterranean Sea.

On the other hand, the $^{14}C$ reservoir ages for this period are exclusively localized over the continental shelf (mainly reconstructed from shallow-water corals and molluscs). These proxy data reveal a high regional variability as reconstructed by Siani et al. (2000) between 1837 and 1951, and Reimer and McCormac (2002), that can be attributed to both: (i) the interactions between the ocean and land by the transport of depleted freshwater, and (ii) the potential changes in the vertical mixing of the water column, with an increase of air-sea $CO_2$ exchanges. These processes could be favoured by the atmospheric conditions, such as the North Atlantic Oscillation (NAO), East Atlantic Pattern (EA), East Atlantic/West Russian pattern (EA/WR) within stronger and frequent wind storms and stronger precipitation over northern Europe (Josey et al., 2011).

After the $^{14}C$ bomb peak, a large gradient of $\triangle^{14}C$ existed between the surface waters already enriched and saturated in bomb $^{14}C$ (values up to 120‰) and intermediate/deep waters with relatively low $\triangle^{14}C$ level (Fig. 5), associated with the long equilibration time with the radiocarbon depleted deep waters, and to vertical mixing. Nevertheless the model simulation shows

that the bomb-produced radiocarbon signal has reached the deep layers of the Mediterranean Sea due to the rapid transfer of surface waters to intermediate and deep depths, especially in the Cretan Sea, where a high $\triangle^{14}C$ is simulated in the deep waters (Fig. 7).

The new $\triangle^{14}C$ data obtained from the analysis of the seawater samples collected during the Meteor M84/3 cruise represent a unique opportunity to critically assess the dynamics of the NEMO-MED12 ocean model and to evaluate its ability to reproduce

the main features of the present day radiocarbon distribution in the Mediterranean. The model produces realistic simulated $\triangle^{14}C$ values in the surface layer that are in agreement with in-situ measurements, thus supporting our modelling approach. However, some important aspects of the model still need to be improved, particularly for deep water, where it underestimates $\triangle^{14}C$ (Fig. 7, Fig. 8). Previous passive tracer evaluations of NEMO-MED12 have shown that the ventilation rates of deep waters are underestimated by the model for the whole Mediterranean (e.g., Ayache et al., 2015a; Palmiéri et al., 2015). This is

particularly evident in the Ionian sub-basin, where the simulated AdDW is too shallow compared to observations, and for the newly-formed WMDW that is not salty enough and its southwards propagation is too slow compared to in-situ observations (e.g., Beuvier et al., 2012a).

Several factors could control the radiocarbon distribution across the Mediterranean Sea. During the pre-industrial period, the AW inflow at Gibraltar, together with freshwater input from rivers could have played an important role on the radiocarbon

distribution in the Mediterranean. The large amount of radiocarbon injected in the atmosphere during the thermonuclear weapon



testing is now the dominant control on the $^{14}C$ distribution in the surface water, completely masking the natural radiocarbon background. This creates the opportunity to study the constraints on the $^{14}CO_2$ air-sea exchange. On the other hand, the ventilation rate is the key mechanism and the most important factor controlling the $^{14}C$ distribution in the deep layer.

The model has provided, for the first time, the evolution of $\triangle^{14}C$ in different parts of the basin and at different depths (Fig. 9). The difference in $\triangle^{14}C$ in surface water between the western and eastern basins reveals enrichment of $\triangle^{14}C$ along surface water-mass pathway due to prolonged exposure of the surface water to atmosphere. It also shows the different mechanism of $^{14}C$ transfer at depth, where it depends on convection processes with higher convection occurring especially during the bomb peak.

The sequence of EMT events that occurred at the beginning of the 1990s in the eastern Mediterranean has substantially changed the deep water-mass structure in the whole basin. Different hypotheses concerning the preconditioning of the EMT and its timing have been proposed in the literature (Roether et al., 2007; Beuvier et al., 2012a; Lascaratos et al., 1999; Theocharis et al., 1999; Klein et al., 1999; Stanev and Peneva, 2002; Josey, 2003). The renewal of the deep water masses after the EMT is satisfyingly simulated by our regional model as illustrated by tritium-helium3 (Ayache et al., 2015a) and by neodymium simulations (Ayache et al., 2016). These findings allow us to study the impact of interannual variability on a very long timescale, including the exceptional events observed in the ventilation of the deep waters. The radiocarbon simulation documents a severe impact of the EMT on the water mass distribution, through the transfer of a large volume of $^{14}C$-enriched near-surface water into the deep layers, with the highest contribution being observed in the area south of the Cretan Arc.

The EMT event generates an important accumulation of $^{14}C$-enriched water at the bottom of Levantine sub-basin with more than 60 ‰ of difference in 1995 compared to the pre-EMT situation. In our simulation the LIW layer is also affected by low values in the eastern Mediterranean, where the renewal of the bottom water masses (low concentration of radiocarbon) during the EMT could lead to a decrease in the $^{14}C$ content in the LIW layer (200-600 m, Fig. 10). On the other hand, higher values of radiocarbon are simulated at intermediate levels in the western Mediterranean during the EMT, with shifts up to 10 ‰ compared to the No-EMT values. During the EMT, part of the Levantine basin is filled by water masses originating in the Aegean Sea, with different characteristics compared to the Adriatic. Hence the EMT could modify water mass characteristics and potentially affect the formation of deep water masses in this basin.

## 5 Conclusions

The radiocarbon distribution of the whole Mediterranean Sea was simulated for the first time using a high-resolution model (NEMO-MED12) at 1/12° horizontal resolution and compared to available in-situ measurements and proxy-based reconstructions. The present study provides a unique opportunity to improve the interpretation/understanding of the available in-situ data, and could help in the design of new observational programmes for the Mediterranean Sea. It also provides a new approach to understanding and better constraining air-sea gas exchange and dynamics of Mediterranean water masses over the last decade. The air-sea exchange parameterization led to a realistic simulation of bomb $^{14}C$ in the surface water, compared to in-situ data. The model correctly simulates the main features of radiocarbon distribution during and after the $^{14}C$ bomb perturbation, es-



pecially in the surface/intermediate layers. On the other hand, severe mismatches between model and observations in the deep layer are clearly associated with shortcomings in the model parameterization.

The natural distribution of $^{14}C$ in the Mediterranean Sea is mainly affected by the inflow of Atlantic water through the Strait of Gibraltar. Further, the large amount of radiocarbon injected into the atmosphere during the nuclear bomb-testing period has been the dominant factor defining the $^{14}C$ distribution in the surface water, largely masking the natural radiocarbon background. More paleo-data from the pre-industrial period would help improve the knowledge of the natural distribution of $^{14}C$ in the Mediterranean and better constrain the fluxes and exchange of radiocarbon between the different reservoirs.

This $^{14}C$ modelling provides a unique opportunity to explore the impact of the interannual variability on the radiocarbon distribution in the whole Mediterranean Sea and the interaction between its western and eastern basins. The outputs of the model simulation of the EMT event reveal a significant increase in $\triangle^{14}C$ (by more than 60 ‰) in the Aegean deep water, and at intermediate level (value up to 10 ‰) in the western basin. The model results with/without EMT show that the vertical transport of surface signals in the Mediterranean is strong, suggesting a major contribution of the EMT in the accumulation of radiocarbon in the eastern Mediterranean deep waters. Although the approach we adopted does not attempt to quantify the anthropogenic carbon, the model results and observations on the $^{14}C$ distribution support the contention that a large amount of anthropogenic carbon is being stored in the deep Mediterranean waters, in agreement with previous findings (e.g., Palmiéri et al., 2015).

## 6 Code availability

The model used in this work is a free surface ocean general circulation model NEMO (Madec and NEMO-Team., 2008) in a regional configuration called NEMO-MED12 (Beuvier et al., 2012a). (http://www.nemo-ocean.eu/)



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





.

**Table 1.** Regional means of radiocarbon reservoir age before 1950 AD. Column 2 gives the observations from Reimer and McCormac (2002); Column 3 the model values in 1940 AD. The uncertainty in the mean is the larger of the standard deviation based on counting statistics and the "standard deviation," which is the square root of the variance.

|  | Age (Yr) | Age (Yr) |
| --- | --- | --- |
| Region | Reimer and McCormac (2002) | Model |
| Western Mediterranean | 400 ± 22 | 402 ± 27 |
| Eastern Mediterranean | 353 ± 47 | 349 ± 14 |
| Algerian sub-basin | 413 ± 51 | 410 ± 27 |
| Tyrrhenian sub-basin | 390 ± 21 | 373 ± 29 |
| Adriatic sub-basin | 396 ± 61 | 349 ± 32 |
| Aegean sub-basin | 480 ± 72 | 336 ± 14 |
| Whole Mediterranean | 400 ± 16 | 379 ± 19 |





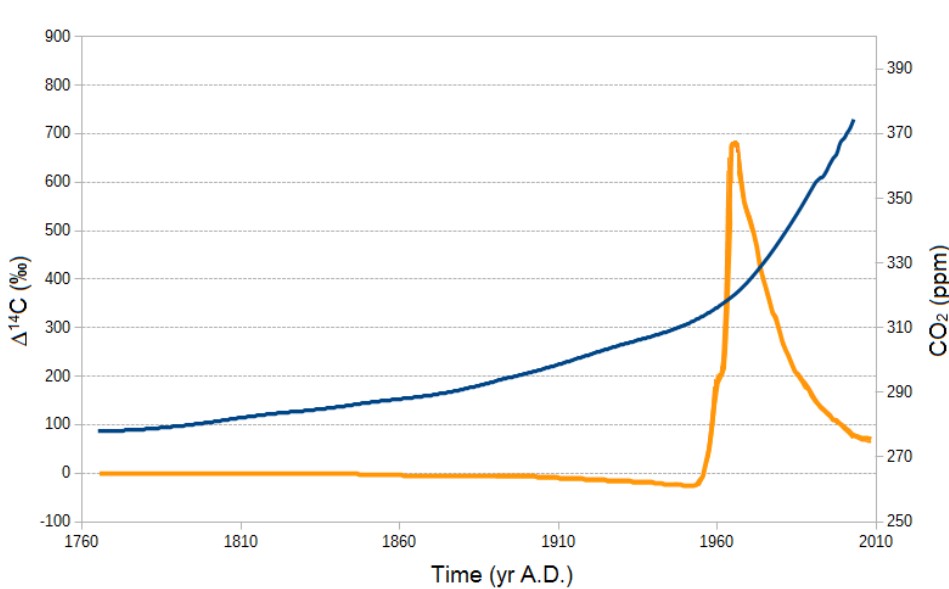

**Figure 1.** Atmospheric $\triangle^{14}C$ in ‰ (orange), atmospheric $CO_2$ in ppm (blue), from Orr et al. (2016) and references cited therein.





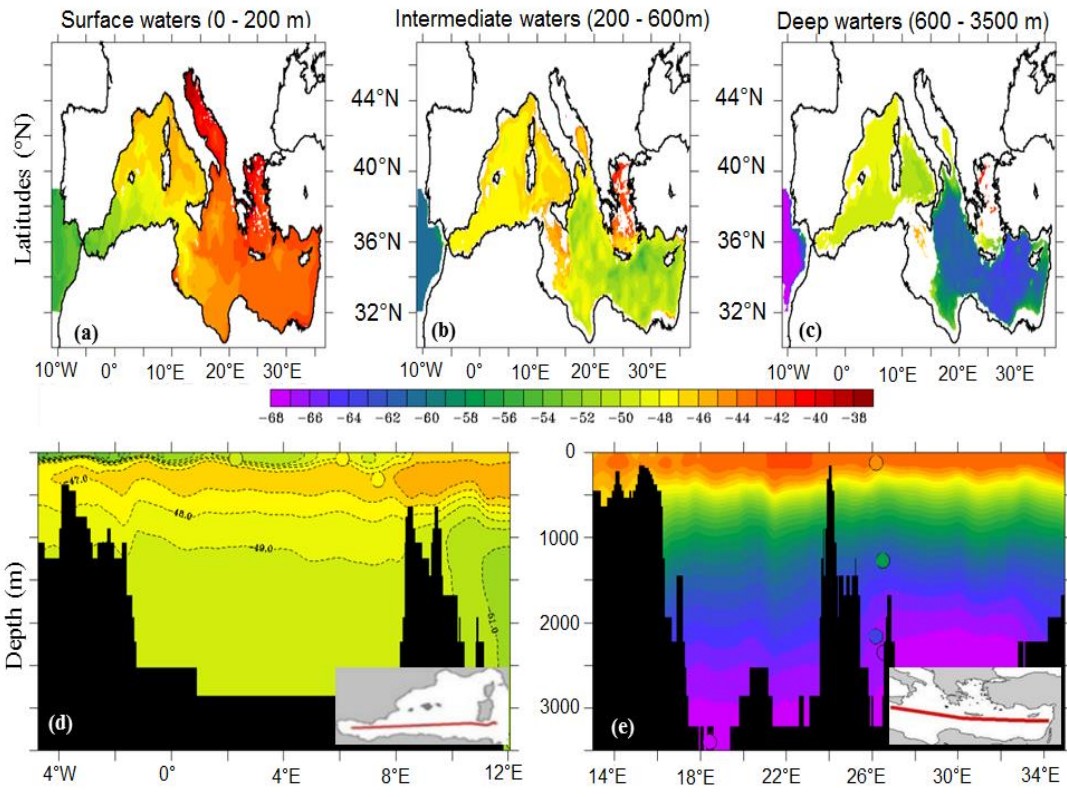

**Figure 2.** Model output for March 1956 showing the pre-bomb situation. Upper panel: mean $\triangle^{14}C$ (in ‰) in surface waters (0 to 200 m), intermediate waters (200 to 600 m), and deep waters (600 to 3500 m). Lower panel: $\triangle^{14}C$ along E-W section in (d) WMed, and (e) EMed, where colour-filled dots represent in-situ observations (Broecker and Gerard, 1969). Both model and data are reported with the same colour scale.





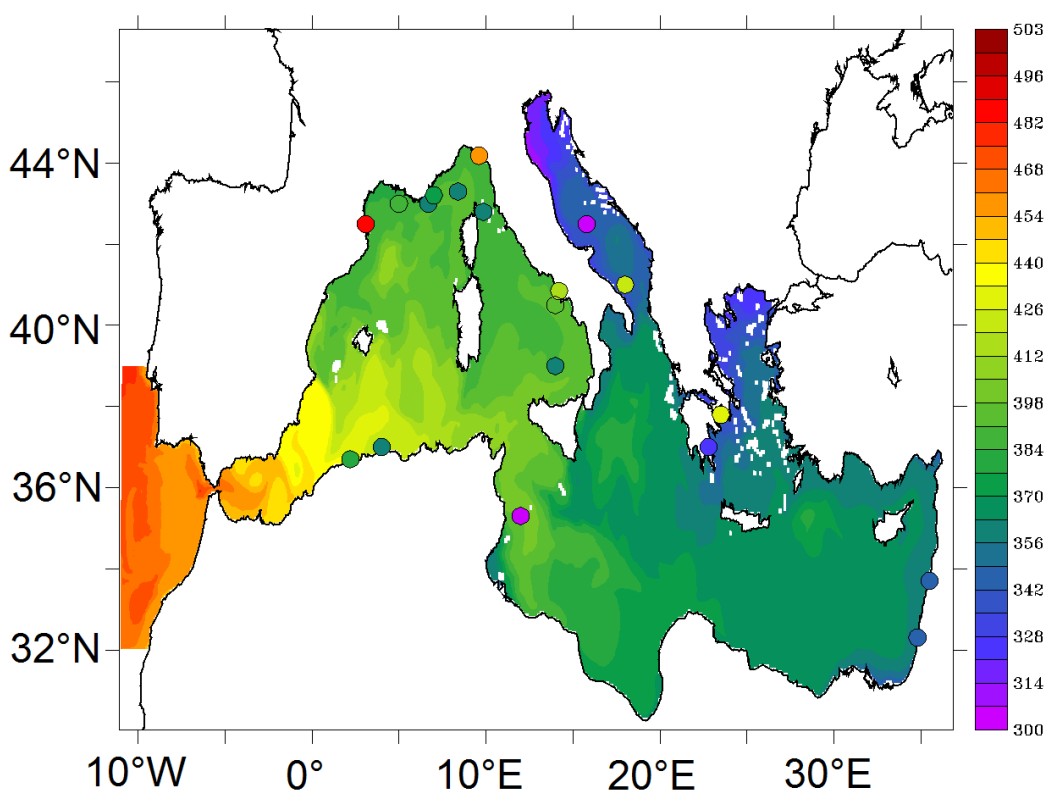

**Figure 3.** Average radiocarbon age (years) in the upper 50 m as computed with the model for 1940. Circles represent reservoir ages derived from measurements of the composition of shells (Siani et al., 2000; Reimer and McCormac, 2002), and from corals (Tisnérat-Laborde et al., 2013).





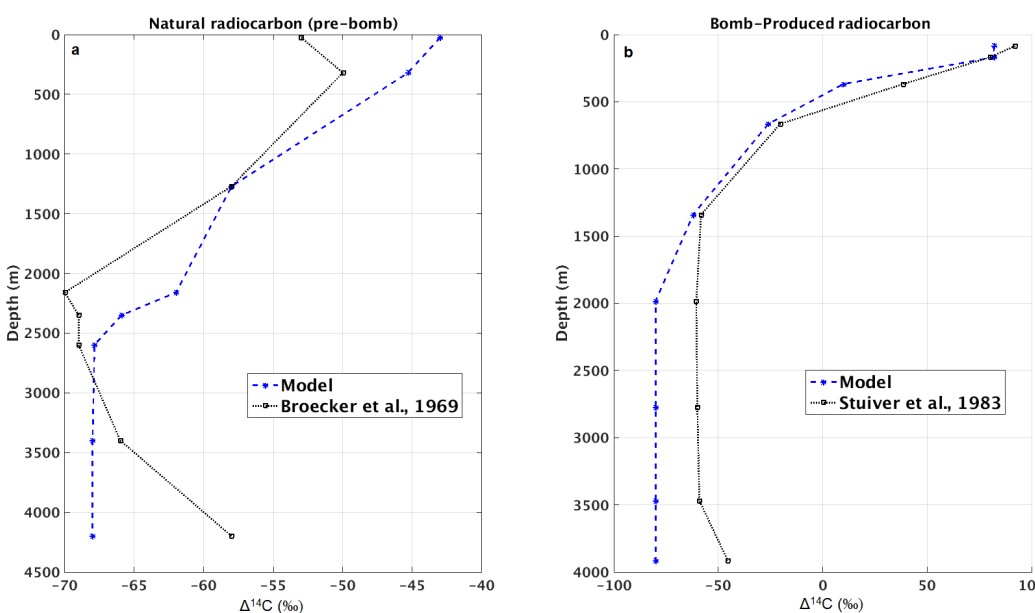

**Figure 4.** Model/data comparison of $\triangle^{14}C$ vertical profiles for (a) the pre-bomb distribution, and (b) total radiocarbon distributions (natural + bomb) in the Eastern basin. Model results are in blue, while black indicates the in-situ data from Broecker and Gerard (1969) and Stuiver et al. (1983).





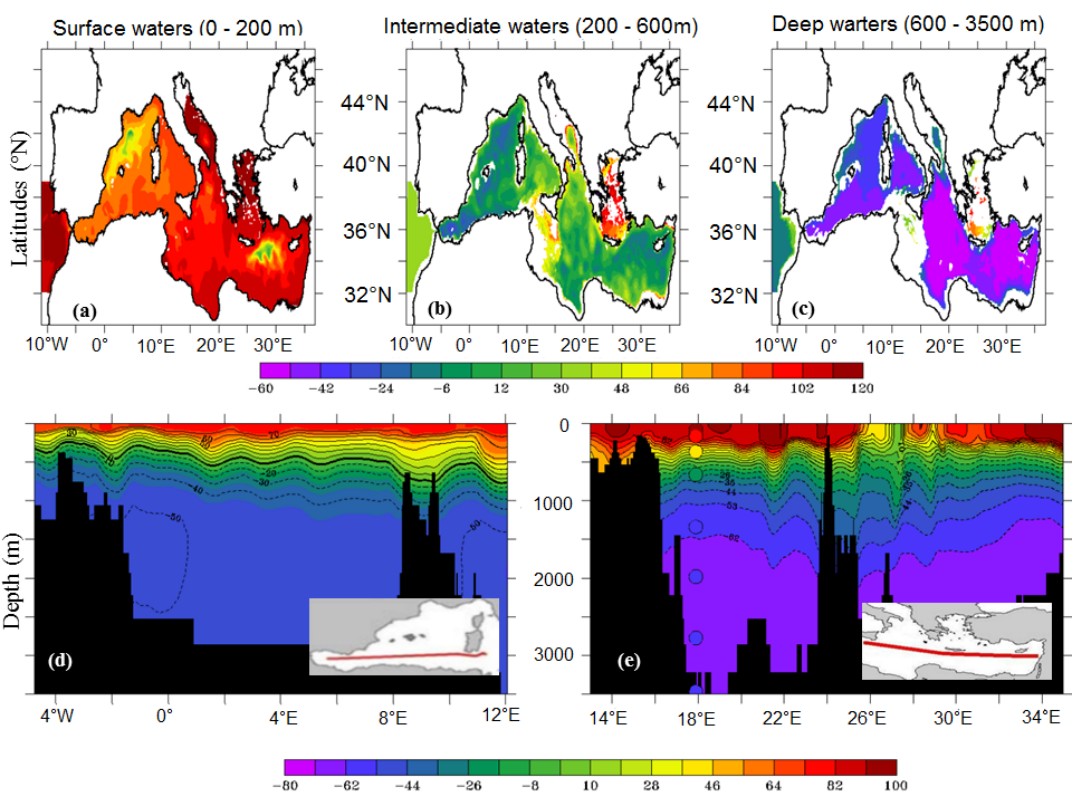

**Figure 5.** Model output for March 1977 for the post-bomb situation. Upper panel: mean $\triangle^{14}C$ (in ‰) in surface waters (0 to 200 m), intermediate waters (200 to 600 m), and deep waters (600 to 3500 m). Lower panel: $\triangle^{14}C$ along E-W section in (d) WMed, and (e) EMed, where colour-filled dots represent in-situ observations (Stuiver et al., 1983). Both model and data are reported with the same colour scale.





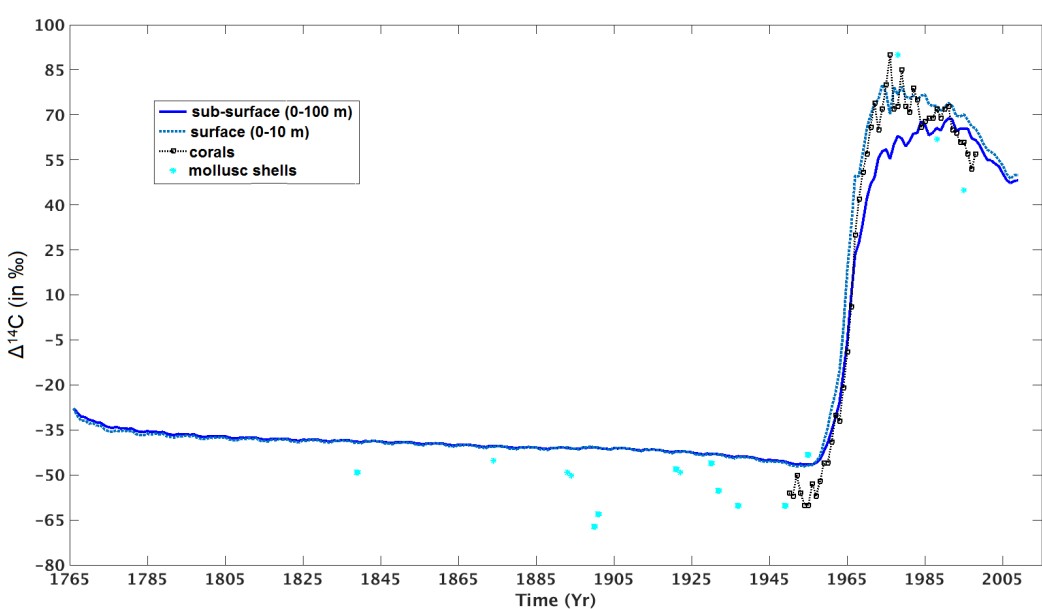

**Figure 6.** $\triangle^{14}C$ values (in ‰) in the Ligurian sub-basin from 1765 to 2008 for the surface water (0-10 m depth; blue dashed line), and sub-surface water (0-100 m depth; blue solid line) together with available in-situ observations (Tisnérat-Laborde et al., 2013) from coral (black dashed line) and molluscs (cyan squares).





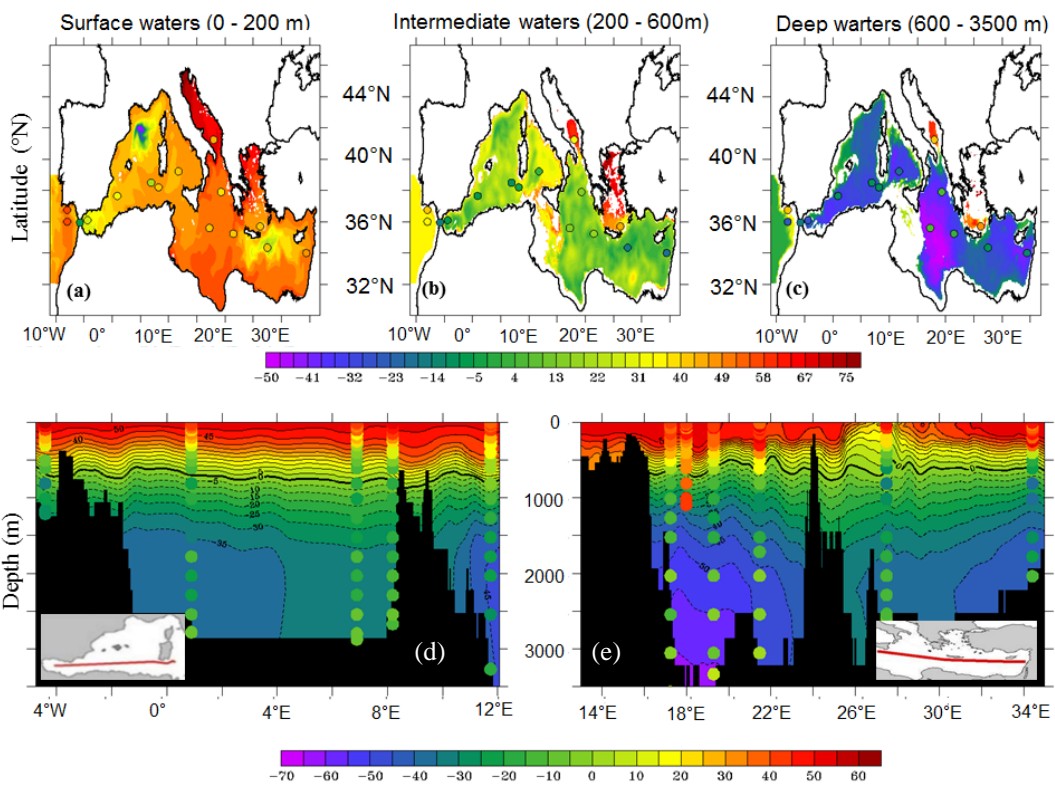

**Figure 7.** Model output in March 2011. Upper panel: mean $\triangle^{14}C$ (in ‰) surface water (0 to 200 m), intermediate water (200 to 600 m), and deep water (600 to 3500 m). Lower panel: $\triangle^{14}C$ along E-W section in (d) WMed, and (e) EMed, where colour-filled dots represent in-situ observations from Meteor M84 Tanhua et al. (2013). Both model and data are reported with the same colour scale.





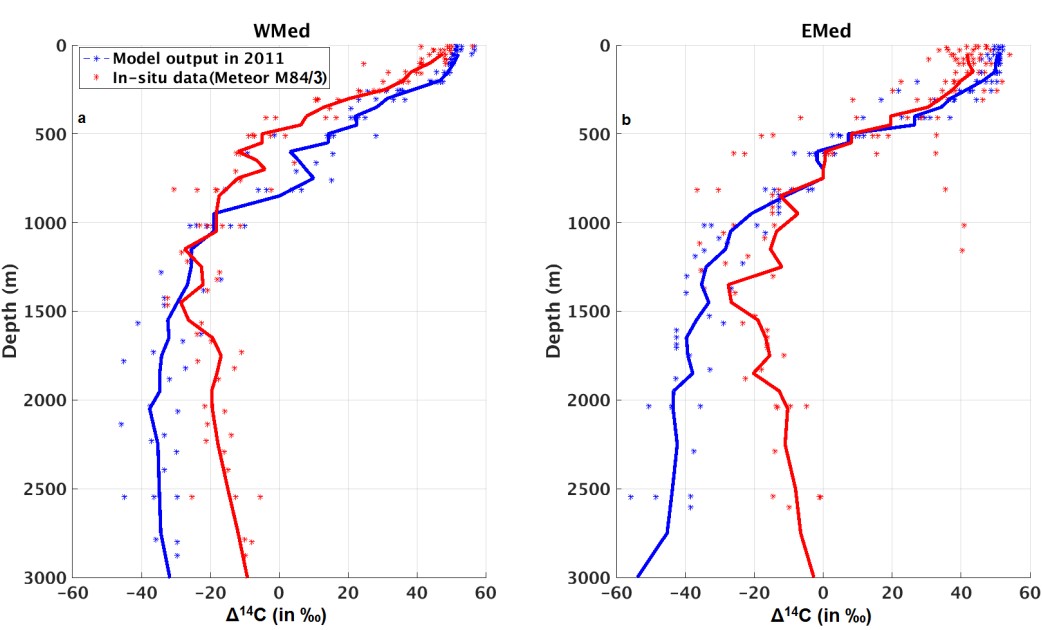

**Figure 8.** Comparison of average vertical profiles of $\triangle^{14}C$ in the WMed (left) and in the EMed (right). Model results are in blue; red indicates the in-situ data



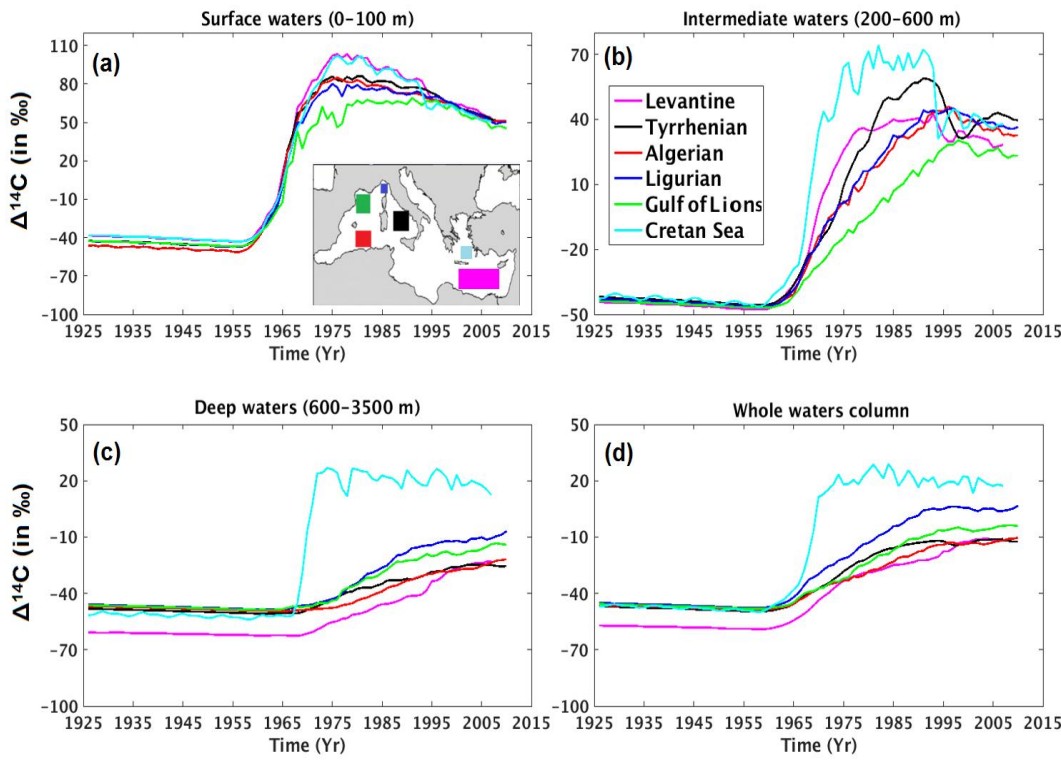

**Figure 9.** $\triangle^{14}C$ evolution from 1925 to 2008, in the Gulf of Lions (green), the Algerian sub-basin (red), the Levantine sub-basin (magenta), the Tyrrhenian sub-basin (black), the Cretan Sea (cyan) and the Ligurian (blue).





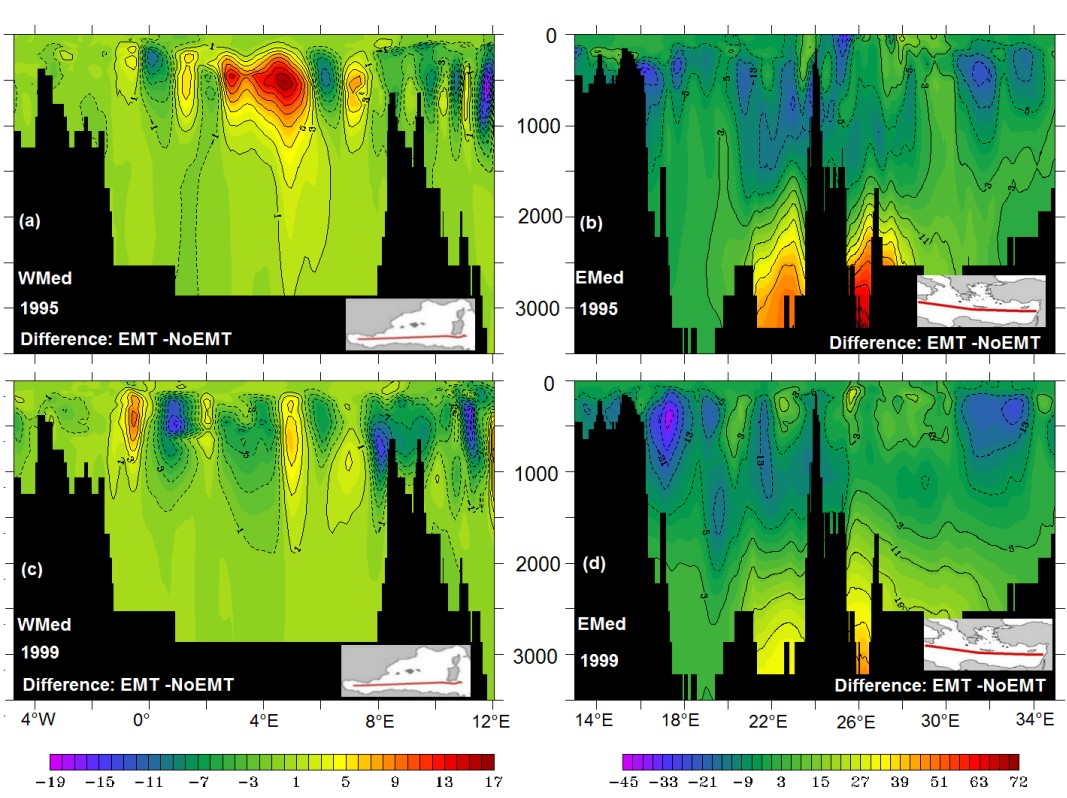

**Figure 10.** $\triangle^{14}C$ difference between EMT and NoEMT experiments along sections in the WMed (left column) and in the EMed (right column) for 1995 (top) and 1999 (bottom).





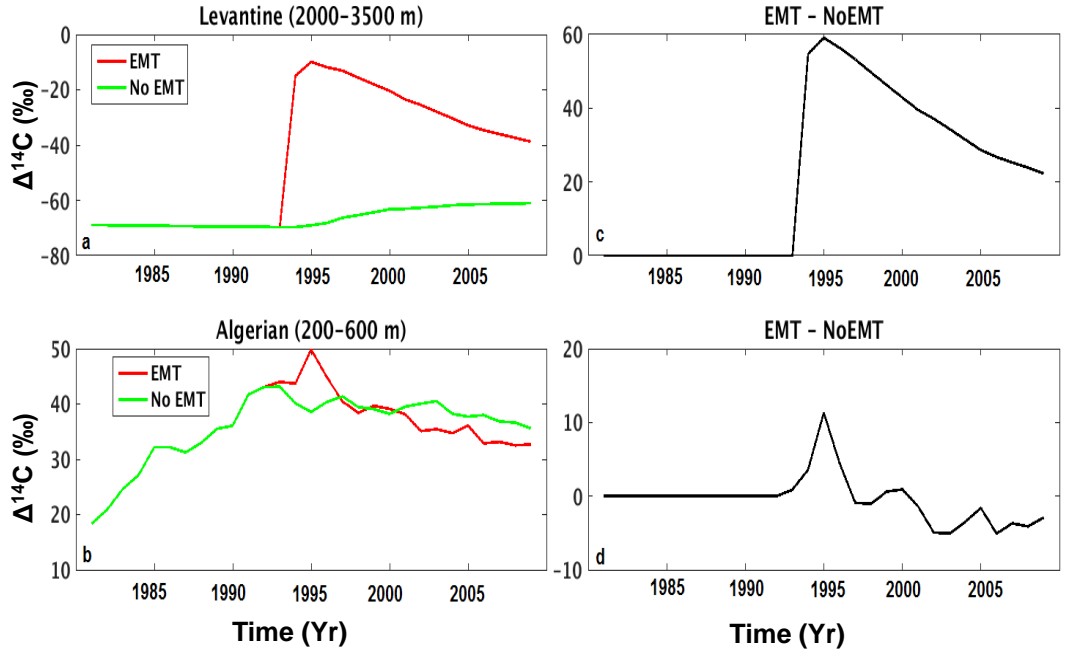

**Figure 11.** Mean $\triangle^{14}C$ obtained for experiment with EMT (red) and NoEMT (green), (a) in the Levantine sub-basin deep water (2000-3500 m depth) and (b) in the Algerian sub-basin at intermediate level (200-600 m). The right panels illustrate the difference between EMT and NoEMT for the corresponding left panels.