# Peer review of "High resolution regional modeling of natural and anthropogenic radiocarbon in the Mediterranean Sea"

_Biogeosciences, 2016_

## Referee Comment (RC1) · Anonymous Referee #1 · 13 Nov 2016

The authors use a high-resolution dynamical model to simulate the distribution of radiocarbon in the Mediterranean Sea. While I feel the topic is relevant and the treatment new, I would like the author would better specify different aspects treated in the paper: a) the role of the Atlantic Water: sensitivity experiments at Gibraltar should be discussed; b) the artificial modifications performed in order to simulate the EMT should be deeply discussed; c) how can convective penetration be increased in the simulations? d) overall, a more critical discussion about limitations of the model simulations should be addressed.

---

## Referee Comment (RC2) · Anonymous Referee #2 · 13 Dec 2016

Review on 'High resolution regional modeling of natural and anthropogenic radiocarbon inthe Mediterranean Sea' by M. Ayache et al.

The paper presents the results of a modeling study of radioactive carbon (14C) for the whole Mediterranean Sea over the pre- and after bomb peak periods. The applied circulation model has been tested before with hydrographic and tracer data (tritium and helium). The model results help to interpret the (sparse) 14C observations in the Mediterranean both from sea water and sediments, and I recommend publication in BG after revision. At some points, I am not quite satisfied with the description/interpretation of the results, and the graphical design of some figures should be improved.

Specific comments:

[Figure]

p.2, l.1-3 'The Mediterranean Sea can be considered as a "miniature ocean", where globalchange can be studied at smaller/shorter spatial and temporal scales (âĹij100 yr compared to more than 1000 yr for the global ocean ...).' The mentioned time scales of 100 vs. 1000 years refer to the overturning time of the Mediterranean/world ocean. Is that really identical with the time scale on which global change is going on, as it is implied by this sentence?

p.2, l.19-22 In this paragraph, 14C is characterized as conservative tracer such as CFCs and tritium. This is not exactly true, as 14C is changed by biology, especially the remineralisation of organic matter. This effect is small and often neglected, but it still is a conceptual difference.

p.3, l.6-8 and p.33, l.31 Here and at some other passages in the paper the role o 14C for the determination of water mass ages and constraining the deep water circulation is mentioned. This is not wrong, but regarding ages, 14C is normally used in older waters with ages of order 1000 yr (comparable to the half life time). For the Mediterranean, tracers with shorter input histories such as CFCs and tritium are more useful. They are also more useful in constraining the deep water pathways in circulation models because the number of observations is much larger than for 14C. This should be made clear somewhere in the text.

p.7, first paragraph on the choice of kw: It seems to me that the choice of kw is the main work regarding the tuning of the circulation model on the base of 14C data. So this topic might be given more room in the description and discussion.

p.8, l.17-18 '... leading to a relatively higher 14 C level in the EMed surface water closer to -46 ‰' Has the value of -46 ‰ a special meaning? Then this should be mentioned in the text. According to Fig. 2a, the values are close to -44 ‰

p.8, l.18-20 'For both western and eastern surface water, the model simulates 14C concentrations slightly higher than the in-situ observations...' I don't see this form the data. In Fig. 2d, 2e and 3, the data are sometimes smaller and sometimes higher than

the model results. The values given in table 1 for model and observations are almost identical for the WMed and EMed, only smaller subregions show significant differences.

p.8, l.20-21 'A careful comparison between model outputs and seawater observations (1959) reveals a more pronounced dis-agreement, especially in the EMed surface water where the model overestimates the 14 C values by more than 10 ‰ (Fig.4a).' Where is the profile shown in Fig. 4a located? Or is it a composite from different locations? If it is one complete profile, the location should be indicated in the inlet map of Fig. 2e or given in coordinates. Second, the measured EMed surface value shown in Fig. 2e is much larger than the value from Fig. 4a, around -45 ‰. So how representative is the profile shown in Fig. 4a for the whole EMed?

p.10, first paragraph Only the higher 14C values in the deepw ater in the Levantine basin are mentioned here, although in the western Med. the values are comparably high between 4°E and 10°E.

p.10, l.13-14 'However the model simulates well the 14 C values in the surface and deep water of Adriatic sub-bassin (Figure 7a and 7c) compared to Meteor M84/3cruise data (Tanhua et al., 2013).' According to Fig. 7a and 7c the model values are too high, which is even more pronounced in Fig. 7b for the intermediate layers.

Figures:

All horizontal maps are strongly distorted. I would prefer a more equidistant representation.

Fig.2: In subfigures b and c, the y-labels have a larger fontsize than the x-labels. The fontsize of the colour bar is too small, and the space between the colour bar and the upper maps should be enhanced.

Fig.3: The font size of the axis labels is too large and of the labels of the color bar too small.

Fig.5: Exactly the same as for Fig.2.

Fig.7: Exactly the same as for Fig.2.

Fig.11: The ylabel 'Time (yr)' should be centered.

Minor comments/corrections:

p.6, Eq.1 the vector 'u' should be notated in bold math

p.9, l.26 '... when we compare...' (not compared)

p.10. l.13 '...values in the surface and deep water of the Adriatic sub-basin' ('the' is missing and 'basin' is misspelled)

p.12, l.3 'However the representation of the pre-bomb distribution is more contrasted in the simulation' I don't understand the meaning of 'contrasted'.

p.13 l.7 '... to prolonged exposure of the surface water to the atmosphere.' (add 'the' before 'atmosphere')

p.13 l.7-8 'where it depends on convection processes with higher convection occurring especially during the bomb peak' I don't see why higher convection has occurred during the bomb peak. Maybe it is meant that the amount of 14C entering the deep water was higher during that time.

p.13, l.18 '... at the bottom of the Levantine sub-basin' ('the' is missing)

---

## Referee Comment (RC3) · Anonymous Referee #3 · 16 Dec 2016

General comments: This work is aimed at providing a basin-scale description of radiocarbon (14C) distribution in the Mediterranean Sea through the NEMO-MED12 dynamical model. Overall, the model seems to mimic the main spatial and temporal variability of 14C from the pre-bomb period until 2008, when the simulation is run. However, some regional patterns are not entirely reproduced, which, in my opinion, could be attributed to the presence of mesoescale phenomena that are not resolved by the model. The model is validated with in-situ measurements and proxy-based reconstructions for the Mediterranean Sea. The authors conclude that Atlantic inflow through the Strait of Gibraltar exerts a high influence on the natural 14C distribution. They also state that by following the propagation of the anthropogenic 14C signal, ventilation of deep Mediterranean waters can be examined, with special emphasis on the effect of the Eastern Mediterranean Transect. Even though data are undoubtedly relevant and novel, I find the work very descriptive and some of the circulation mechanisms behind 14C patterns are not fully described (or even mentioned). At a certain point, it seems that the study is only focused on showing the robustness of the modelling approach, in other words, how well simulations mimic the available 14C records rather than providing a view of how circulation is responsible of radiocarbon evolution and patterns in the basin before and after the atmospheric bomb tests. I miss, for instance, comparisons with other ocean regions, or a more comprehensive discussion (explanations) on the discrepancies found between the model output and the in situ data. Potential behaviour of radiocarbon under other scenarios in the basin would have been also nice.

Nevertheless, taking into account its novelty, this work deserves publication in Biogeosciences although revisions and modifications should be introduced in the manuscript in order to clarify some of the conclusions drawn by the authors.

Specific comments:

Abstract

The simulation was run until 2010 to give the post-bomb distribution.

I believe the simulation is run until 2008 although model outputs are compared with in situ measurements taken in 2011.

Introduction Page 1, Lines 15-21: The whole paragraph seems to be out of scope. I do not see the relationship between the stresses suffered by the MedSea and the distribution of radiocarbon

Page 2, Lines 6-10. The excess of evaporation versus precipitation does not transform Atlantic waters into Mediterranean waters and leads AW to sink offshore. It is actually the process that drives the entry of Atlantic waters through the Strait of Gibraltar to compensate water loss and keep the mass balance. Water masses formation in the

basin are subsequently the result of other phenomena more related with atmospheric forcing and density gradients. It might be a small detail but conceptually it is important, especially for readers not familiar with the MedSea. I would suggest to re-write the paragraph.

Page 2, Line 21. I would not compare CFC and tritium with 14C, as this one is not entirely passive. I understand what the authors mean by the sentence but radiocarbon is indeed used by the biological community so I would recommend to state that circumstance or simply not to equate the tracers.

Results

Page 8, lines 17-18: According to Figs 2 and 3, the model does not overestimate the radiocarbon concentration in surface everywhere in the basin but it depends on the particular region. Also, plots in Fig. 2 could be manifestly enhanced as it is hard to distinguish in situ observations over the contour in the graphs.

Page 8, Line 20. The careful comparison between vertical profiles of model outputs and seawater observations in Fig. 4 is restricted to the Eastern basin. Why the Western basin is not considered if according to Fig 3 there are also some disagreements? Not enough in situ data to compare? Please clarify

Page 8, Line 27. Any idea why the pre-bomb radiocarbon levels differ so much between the in situ data and the model outputs in the Aegen sub-basin (Table 1)? I guess there must be some circulation patterns not resolved in the model. Plus, I do not understand the sentence the range in the observations is also high.

Page 9, Line 10. At depth, the model tends to underestimate the 14C penetration in the deep Ionian sub-basin, where it fails to reproduce the high 14C levels associated with EMDW formation (Fig. 4b). Where is this disagreement shown in Fig. 4b? Does the plot correspond to that particular sub-basin or to the entire Eastern basin?

Page 9, Line 24. The greater is the mixing layer depth, the weaker is the amplitude and

the peak is delayed. Is this sentence grammatically correct?

Page 9 and 10: To me, Fig. 7 depicts too much disagreement between simulated distributions and in situ data in many regions, not only in deep convection areas. For instance, even though it is hard to see the symbols in Fig. 7a, it seems that in surface waters of the Strait of Gibraltar the model overestimates the radiocarbon concentration by more than 20 ‰.Plus, in the discussion section it is stated that there is no time series data of 14C concentration in that area, while in the graph there are at least 4 measurements in the gulf of Cadiz and within the channel of the Strait. Could have they been used to fuel the model? In addition, explanation of data indicated in Figs 7 and 8 is confused, as description of patterns jumps from one to another without a logic sequence.

Page 10: The radiocarbon time evolution spans from 1925 to 2008, why this particular year? In fact, Fig. 7 shows comparison between the model outputs and data of the 2011 Meteor cruise, which included measurements throughout the whole basin. Why the simulated evolution does not run until then? It would be interesting to confirm that evolution follows the pattern indicated in Fig. 7. Also, I would keep the same vertical scale in all plots to facilitate comparisons. The response found in intermediate-deep waters of the gulf of Lions is somehow unexpected, as deep convection events during winter should favor the sink of radiocarbon, particularly in extreme winters, such as that occurring in the area in 2004/2005. In Fig 9d, the intermediate layer of the gulf of Lions exhibit the lowest radiocarbon levels after the bomb episodes and deeper waters are characterized by values even lower than those found in the Tyrrhenian sub-basin. Is there any explanation for that? Moreover, are data in plot 9d integrated values through the whole water column? These results are not explained in the text. Plus, the title is wrong, it should say whole water columns.

Discussion

Page 11, Line 10. The radiocarbon simulations provide independent and additional

constraints on the thermohaline circulation and deep-water ventilation in the Mediterranean Sea. I do not see this in the manuscript. It would be the other way around. Data are interpreted according to the general circulation mechanisms known to proceed in the Med Sea.

Page 12. The comparison between the model outputs and the 14C values from in-situ data reported by Broecker and Gerard (1969), Stuiver et al. (1983) and Tanhua et al. (2013) reveals a good model performance in simulating the bomb/post-bomb radiocarbon distribution (Fig. 4b, Fig. 8). However the representation of the pre-bomb distribution is more contrasted in the simulation (Fig. 4a). I do not understand this paragraph. In fact, those two figures in particular show the largest disagreements, particularly in intermediate-deep waters and for the bomb-produced radiocarbon.

Page 13, Line 7: with higher convection occurring especially during the bomb peak. Where is this shown in the paper?

Conclusions The natural distribution of 14C in the Mediterranean Sea is mainly affected by the inflow of Atlantic water through the Strait of Gibraltar

As far as I understood, the concentration of radiocarbon in the Atlantic inflow did not come from in situ data or available measurements since it was taken from previous modeling approaches (as indicated in different sections of the paper). Therefore, this study does not show per se, the influence of the Atlantic radiocarbon on the distribution of this tracer in the Med Sea, as it is a fixed value used to fuel the model. The paper actually demonstrates that the entry of Atlantic waters is essential for water masses formation and circulation in the Mediterranean, which is a very well known topic and which, in turn, regulates the distribution of radiocarbon. In fact, it would have been interesting to perform the same simulations by changing for instance the values of the water masses transport through the Strait or the radiocarbon concentration associated to the Atlantic jet. To me, such conclusion cannot be drawn from the data. I would omit it here and in the abstract or at least, the sentence should be re-written.

---

## Author Comment (AC1) · 13 Feb 2017

**We thank the reviewer #1 for her/his constructive comments on the manuscript. We have carefully considered all questions and concerns raised. The structure of our reply is as follows; each comment from the anonymous reviewer is recalled in blue, and our reply in black.**

The authors use a high-resolution dynamical model to simulate the distribution of radiocarbon in the Mediterranean Sea. While I feel the topic is relevant and the treatment new, I would like the author would better specify different aspects treated in the paper:

a) the role of the Atlantic Water: sensitivity experiments at Gibraltar should be discussed;

We have performed two simulations with different boundary conditions at Gibraltar (red and blue boxes and lines, see Fig.1, below); the first time series (red box) gives very low level of radiocarbon in the Mediterranean Sea (as represented in Fig.2, below). In the second simulation we used a larger box (blue in Fig.1), where results are more realistic compared to some data from the North Atlantic (Tisnérat-Laborde et al., 2013; Tisnerat-Laborde, personal communication). The radiocarbon simulation greatly improves when using the larger box as boundary conditions, hence this was used to simulate $^{14}$C in the Mediterranean Sea.

This part has been clarified in the revised version of the manuscript.

**[See changes p 7, line 18-25 in the revised manuscript.]**

[Figure]

Fig. 1: The concentration of radiocarbon in the Atlantic inflow (NEMO global model, Mouchet et al. 2016).

[Figure]

Fig.2: $\Delta^{14}C$ values (in ‰) in the Ligurian sub-basin from 1765 to 2008 for the surface water (0-10 m depth; blue), together with available in-situ observations (Tisnérat-Laborde et al., 2013) from coral (black dashed line). Simulated data obtained using the smaller box (Fig. 1) as boundary conditions.

b) the artificial modifications performed in order to simulate the EMT should be deeply discussed;

To improve dense-water fluxes through the Cretan Arc during the EMT (1992-1993) the ARPERA forcings were modified over the Aegean sub-basin (Beuvier et al., 2012a), by increasing mean values as done by Herrmann and Somot (2008) for the Gulf of Lions. More specifically, from November to March for the winters 1991-1992 and 1992-1993, daily surface heat loss was increased by 40 W $m^{-2}$, daily water loss by 1.5 mm and the daily wind stress modulus by 0.02 N $m^{-2}$. These changes accelerate the transfer of surface temperature and salinity perturbations into intermediate and deep layers of the Aegean subbasin, and improve the dense-water formation in the Aegean sub-basin during the EMT, with more intense mixing from winter convection.

The artificial modifications performed in order to simulate the EMT were fully discussed by Beuvier et al (2010), and in our previous work on anthropogenic tritium modelling (Ayache et al., 2015a). In this study we have used the same parametrization and method as in those previous work, therefore we do not think it is necessary to deeply discuss the details in the present manuscript.

**[See changes, p 07 line 30-35 in the revised manuscript.]**

c) how can convective penetration be increased in the simulations?

The convective penetration is more important in the classical area of deep convection in the Mediterranean Sea (i.e. Gulf of Lion, Adriatic and the Aegean sub-basins…) where the surface heat loss, water loss and the wind stress are more important on those areas.

In this work we have used a high resolution dynamical model (NEMO-MED12, Beuvier et al. 2012) based on the tagged version nemo v3.2 of the NEMO ocean general circulation model (Madec et al., 2008). This model was only forced by the atmospheric model AREPERA and we prescribe the initial and boundary conditions (as detailed in the manuscript section 2.1). Increasing the convective penetration could be obtained by changing air sea fluxes or in the Adriatic changing river runoff, but it is not the goal of this paper.

d) overall, a more critical discussion about limitations of the model simulations should be addressed.

We agree with the reviewer that limitations of the model simulations should be more critically discussed.

Previous passive tracer evaluations of NEMO-MED12 (e.g., Ayache et al., 2016; Ayache et al., 2015a; Palmiéri et al., 2015) have shown that the model satisfactorily simulates the main structures of the thermohaline circulation of the Mediterranean Sea, with mechanisms having a realistic timescale compared to observations.

However tritium/helium-3 simulations from Ayache et al. (2015) have highlighted the too-weak formation of Adriatic Deep Water (AdDW), followed by a weak contribution to the Eastern Mediterranean Deep Water (EMDW) in the Ionian sub-basin. In the western basin, the production of WMDW is correctly simulated, but the spreading of the recently ventilated deep water to the south of the basin is too weak. The consequences of these weaknesses in the model's skill in simulating some important aspects of the dynamics of the deep ventilation of the Mediterranean will have to be kept in mind when analyzing the model output.

We thank the reviewer for this suggestion. This part has been extended in the revised version.

**[See changes p 12 line 34, in the revised manuscript.]**

---

## Author Comment (AC2) · 13 Feb 2017

**We thank reviewer #2 for her/his constructive comments on the manuscript. We have carefully considered all questions and concerns raised. The structure of our reply is as follows; each comment from the anonymous reviewer is recalled in blue, and our reply in black.**

Specific comments:

p.2, l.1-3 'The Mediterranean Sea can be considered as a "miniature ocean", where global change can be studied at smaller/shorter spatial and temporal scales (â´Lij100 yr compared to more than 1000 yr for the global ocean ...).' The mentioned time scales of 100 vs. 1000 years refer to the overturning time of the Mediterranean/world ocean. Is that really identical with the time scale on which global change is going on, as it is implied by this sentence?

In many study the Mediterranean Sea is described as a miniature ocean (e.g. Lascaratos et al 1999) based on the difference of the overturning time of the Mediterranean/world ocean. Most of the physical processes that characterize the global general ocean circulation (e.g. intermediate and deep water formation) also occur in the Mediterranean Sea but at shorter time scale. This allows investigating human-induced climate modifications that are rapidly transferred to sub-surface waters in the entire Mediterranean Sea. For example, the increase of seawater temperature at intermediate and deep level due to the effect of the present global warming is stronger in the Mediterranean Sea compared to that observed at similar depths in the global ocean. Similarly, acidification due to uptake of anthropogenic carbon is already affecting all deep water masses of the Mediterranean Sea (Palmieri et al, 2015) Owing to its small size and limited exchange with the Atlantic Ocean, the Mediterranean Sea amplifies the effects of global changes, which can be then studied at shorter temporal scales.

The shorter Mediterranean turnover timescales permits to perform longer and more computational efficient simulations.

For the sake of clarity, we have modified this sentence in the revised version of the manuscript.

**[See changes p 2, line 2-4 in the revised manuscript.]**

p.2, l.19-22 In this paragraph, $^{14}$C is characterized as conservative tracer such as CFCs and tritium. This is not exactly true, as $^{14}$C is changed by biology, especially the remineralisation of organic matter. This effect is small and often neglected, but it still is a conceptual difference.

We thank the referee for this suggestion; this conceptual difference between $^{14}$C and the other tracers has been clarified in the revised manuscript.

**[See changes p 2, line 25-26 in the revised manuscript.]**

p.3, l.6-8 and p.33, l.31 Here and at some other passages in the paper the role of 14C for the determination of water mass ages and constraining the deep water circulation is mentioned. This is not wrong, but regarding ages, 14C is normally used in older waters with ages of order 1000 yr (comparable to the half-life time). For the Mediterranean, tracers with shorter input histories such as CFCs and tritium are more useful. They are also more useful in constraining the deep water pathways in circulation models because the number of observations is much larger than for 14C. This should be made clear somewhere in the text.

We agree with the referee, that tracers with shorter input histories are more adapted to investigate water mass ages and circulation in the Mediterranean Sea (see for example Ayache et al., 2015a for anthropogenic tritium and Palmieri et al., 2015 for CFCs).

The present radiocarbon simulation aims at implementing a geochemical tracer with a longer time scale allowing more paleo-oriented applications. This $^{14}$C modelling would help improving the knowledge of the natural distribution of $^{14}$C in the Mediterranean, providing a unique opportunity to explore the impact of the interannual/decadal variability on radiocarbon distribution in the Med Sea.

Clarified in revised version

**[See changes p 4, line 19-21 in the revised manuscript.]**

p.7, first paragraph on the choice of kw: It seems to me that the choice of kw is the main work regarding the tuning of the circulation model on the base of 14C data. So this topic might be given more room in the description and discussion.

For the Mediterranean Sea we have studied the impact of Kw on the radiocarbon distribution in this semi-enclosed basin, and we have chosen a value that gives the best agreement with available in-situ data. On the other hand the present simulation was done in a computationally efficient off-line mode (as mentioned in section 2.2), i.e. the dynamic was run independently from the $^{14}$C module and the Kw parametrization was adapted for the Mediterranean Sea.

A sentence was added to clarify this point.
**[See changes p 7, line 4-7 in the revised manuscript.]**

p.8, l.17-18 '... leading to a relatively higher $^{14}$C level in the EMed surface water closer to -46 ‰·' Has the value of -46 ‰ a special meaning? Then this should be mentioned in the text. According to Fig. 2a, the values are close to -44 ‰

The referee is correct, the value is closer to -44 and this has been corrected in the revised version. The -46 ‰ has no special meaning.

**[See changes p 8, line 20 in the revised manuscript.]**

p.8, l.18-20 'For both western and eastern surface water, the model simulates 14C concentrations slightly higher than the in-situ observations...' I don't see this form the data. In Fig. 2d, 2e and 3, the data are sometimes smaller and sometimes higher than the model results. The values given in table 1 for model and observations are almost identical for the WMed and EMed, only smaller subregions show significant differences.

The referee is correct, there are no significant differences between WMed and the EMed average values. However, if we look at Fig. 3 (data from Siani et al. 2000) there is an important spatial gradient across the different sub-basins in the Mediterranean Sea as a consequence of old carbon impact near the coastal areas. This effect is not represented in the present simulation. On the other hand, our model results are in good agreement with average values provided by

Reimer and McCormac (2002) for each sub basin in the Mediterranean. Other in-situ data would help to improve the model parametrization.

p.8, l.20-21 'A careful comparison between model outputs and seawater observations (1959) reveals a more pronounced dis-agreement, especially in the EMed surface water where the model overestimates the 14C values by more than 10‰ (Fig.4a).' Where is the profile shown in Fig. 4a located? Or is it a composite from different locations? If it is one complete profile, the location should be indicated in the inlet map of Fig. 2e or given in coordinates. Second, the measured EMed surface value shown in Fig. 2e is much larger than the value from Fig. 4a, around -45 ‰ So how representative is the profile shown in Fig. 4a for the whole EMed?

The vertical profile shown in Fig. 4a is a composite of seawater observations from different locations (Brocker et al., 1969) as represented in Fig. 2 for the pre-bomb situation. The measured EMed surface value shown in Fig. 2e is much larger than the value from Fig. 4a because the latter presents the average value of all in-situ data and model output on the same position. However the representation of the pre-bomb distribution is more contrasted in the simulation, where several issues complicate the simulation of the natural steady state distribution of $^{14}$C using ocean-model circulation (e.g. the uncertainty associated with the boundary conditions).
For the sake of clarity, we have modified Fig.4 caption in the revised manuscript.

p.10, first paragraph Only the higher 14C values in the deep water in the Levantine basin are mentioned here, although in the western Med. the values are comparably high between 4 °E and 10 °E.

Added in the revised manuscript.
**[See changes p 10, line 11 in the revised manuscript.]**

p.10, l.13-14 'However the model simulates well the 14 C values in the surface and deep water of Adriatic sub-bassin (Figure 7a and 7c) compared to Meteor M84/3cruise data (Tanhua et al., 2013).' According to Fig. 7a and 7c the model values are too high, which is even more pronounced in Fig. 7b for the intermediate layers?

We agree with the referee that the model values in the Adriatic deep water are higher compared to those obtained for the Meteor M84/3 cruise. However the high $^{14}$C level in the deep water proved that the model simulates deep convection in the Adriatic sub-basin. Nevertheless, the outflow of this deep water through the Strait of Otranto is weaker in the model and the simulated signal of deep-water ventilation from the Adriatic sub-basin is propagating at too shallow depth compared to the observations. This shortcoming was also noticed for the other tracer simulations with the same model NEMO-MED12 (e.g. Ayache et al. 2015a; Palmiéri et al., 2015; Ayache et al. 2015b; Ayache et al. 2016). In the Adriatic sub-basin, the contribution of rivers is very important; however, the atmospheric forcing ARPERA combined with the river runoff data set overestimates the freshwater flux, and provides too much freshwater on this domain. This results in unusually low-salinity water compared to observations, preventing winter convection and the propagation of AdDW to the bottom of the Ionian sub-basin.

Figures:

Fig.2: In subfigures b and c, the y-labels have a larger fontsize than the x-labels. The fontsize of the colour bar is too small, and the space between the colour bar and the upper maps should be enhanced.

Adjusted

Fig.3: The font size of the axis labels is too large and of the labels of the color bar too small.

Corrected

Fig.5: Exactly the same as for Fig.2.

Adjusted

Fig.7: Exactly the same as for Fig.2.

Adjusted

Fig.11: The ylabel 'Time (yr)' should be centered.

Corrected

Minor comments/corrections:

p.6, Eq.1 the vector 'u' should be notated in bold math

Corrected

p.9, l.26 '... when we compare...' (not compared)

Corrected

p.10. l.13 '...values in the surface and deep water of the Adriatic sub-basin' ('the' is missing and 'basin' is misspelled)

Corrected

p.12, l.3 'However the representation of the pre-bomb distribution is more contrasted in the simulation' I don't understand the meaning of 'contrasted'.

Replaced by 'more difficult"

p.13 l.7 '... to prolonged exposure of the surface water to the atmosphere.' (add 'the' before 'atmosphere').

Done

p.13 l.7-8 'where it depends on convection processes with higher convection occurring especially during the bomb peak' I don't see why higher convection has occurred during the bomb peak. Maybe it is meant that the amount of 14C entering the deep water was higher during that time.
The transfer of radiocarbon was higher during the bomb peak as a consequence of large amount of $^{14}$C in the atmosphere. We agree with the referee that this sentence is not clear and it can be easily misinterpreted. It has been modified in the revised version.

**[See changes p 13, line 16-17 in the revised manuscript.]**

p.13, l.18 '... at the bottom of the Levantine sub-basin' ('the' is missing)

Corrected.

---

## Author Comment (AC3) · 13 Feb 2017

**We thank reviewer #3 for her/his constructive comments on the manuscript. We have carefully considered all questions and concerns raised. The structure of our reply is as follows; each comment from the anonymous reviewer is recalled in blue, and our reply in black.**

Specific comments:

Abstract

The simulation was run until 2010 to give the post-bomb distribution. I believe the simulation is run until 2008 although model outputs are compared with in situ measurements taken in 2011.

The $^{14}$C simulation was done in a computationally efficient off-line mode, which allows us to run simulation of different tracers (e.g. Palmieri et al 2015; Ayache et al. 2015), in pre-computed transport fields instead of re-computing them, which is very costly.
The dynamical fields are available until 2011 from Beuvier et al. (2012). Starting from the end of the pre-industrial equilibrium run, the model was integrated from 1765 to 2011 (as mentioned page 7 line 13) covering the Suess effect (SUESS, 1955), the entire radiocarbon transient generated by the atmospheric nuclear weapon tests performed in the 1950s and early 1960s as well as the anthropogenic $CO_2$ increase.

Corrected in the revised manuscript
**[See changes p 1, line 4 in the revised manuscript.]**

Introduction Page 1, Lines 15-21: The whole paragraph seems to be out of scope. I do not see the relationship between the stresses suffered by the MedSea and the distribution of radiocarbon

In addition to providing constraints on radiocarbon distribution in the Mediterranean, our simulations provide information on the ventilation of the deep Mediterranean waters which is useful for assessing NEMO-MED12 performance.
This study is part of the work carried out to assess the robustness of the NEMO-MED12 model, which will be used to study the evolution of the climate and its effect on the biogeochemical cycles in the Mediterranean Sea, and to improve our ability to predict the future evolution of the Mediterranean Sea under the increasing anthropogenic pressure (e.g. Drobinski et al., 2012; Beuvier et al., 2010; Herrmann et al., 2010; Somot et al., 2006).

Page 2, Lines 6-10. The excess of evaporation versus precipitation does not transform Atlantic waters into Mediterranean waters and leads AW to sink offshore. It is actually the process that drives the entry of Atlantic waters through the Strait of Gibraltar to compensate water loss and keep the mass balance. Water masses formation in the basin are subsequently the result of other phenomena more related with atmospheric forcing and density gradients. It might be a small detail but conceptually it is important, especially for readers not familiar with the MedSea. I would suggest to re-write the paragraph.

We thank the referee for this suggestion, this paragraph is rephrased in the revised version according to reviewer's comments.

**[See changes p 2, line 7-10 in the revised manuscript.]**

Page 2, Line 21. I would not compare CFC and tritium with $^{14}$C, as this one is not entirely passive. I understand what the authors mean by the sentence but radiocarbon is indeed used by the biological community so I would recommend to state that circumstance or simply not to equate the tracers.

This point has been also raised by reviewer #2 and it has been addressed in the revised version of the manuscript.

Results
Page 8, lines 17-18: According to Figs 2 and 3, the model does not overestimate the radiocarbon concentration in surface everywhere in the basin but it depends on the particular region. Also, plots in Fig. 2 could be manifestly enhanced as it is hard to distinguish in situ observations over the contour in the graphs.

The number of in-situ data for the pre-industrial period is very limited in the Mediterranean Sea. Figure 2 shows that the east-west gradient was satisfactorily captured by the model, with a slight overestimation of $^{14}$C concentrations in the surface water of western basin. However results on figure 3 are reservoir ages and as discussed in the manuscript they are influenced by other sources (e.g. coastal input of "old carbon") so that only the spatial structure is discussed (east-west gradient).
More paleo-data from the pre-industrial period would help improving the knowledge of the natural distribution of $^{14}$C in the Mediterranean.

Fig.2 was improved as suggested by the reviewer.

Page 8, Line 20. The careful comparison between vertical profiles of model outputs and seawater observations in Fig. 4 is restricted to the Eastern basin. Why the Western basin is not considered if according to Fig 3 there are also some disagreements? Not enough in situ data to compare? Please clarify

The careful comparison between vertical profiles of model outputs and in-situ data in Fig.4 is restricted to the EMed because there is no data available for the WMed for the bomb situation from Stuiver et al. (1983).
Fig. 3 presents only the surface values from paleo-reconstruction, and there is no data for the deep waters.

Page 8, Line 27. Any idea why the pre-bomb radiocarbon levels differ so much between the in situ data and the model outputs in the Aegen sub-basin (Table 1)? I guess there must be some circulation patterns not resolved in the model. Plus, I do not understand the sentence the range in the observations is also high.

The simulation of natural radiocarbon is particularly difficult because the average dynamical circulation used in the present study does not produce enough convection to pull-up the old carbon accumulated in the deep water. In addition, our simulation does not take into account the potential impact of old carbon in the coastal area, which could be the case in the Aegean sub-basin.

The sentence 'the range in the observations is also high' means that the range of uncertainty is higher in the observations.
Clarified in revised version.
**[See changes p 8, line 30 in the revised manuscript.]**

Page 9, Line 10. At depth, the model tends to underestimate the $^{14}$C penetration in the deep Ionian sub-basin, where it fails to reproduce the high $^{14}$C levels associated with EMDW formation (Fig. 4b). Where is this disagreement shown in Fig. 4b? Does the plot correspond to that particular sub-basin or to the entire Eastern basin?

The vertical profile potted in Fig.4b represents the model result in the Ionian sub-basin together with in-situ data measured by Stuiver et al. 1983 at 18 °E.

As mentioned in the Introduction, in this study we used the NEMO-Med12 dynamical model that was already tested and evaluated with other tracers, such as tritium (Ayache et al., 2015a), helium (Ayache et al., 2015b) and CFC (Palmieri et al., 2015). Those tracers have highlighted that the model simulates a too-weak formation of Adriatic Deep Water (AdDW), followed by a weak contribution to the Eastern Mediterranean Deep Water (EMDW) in the Ionian sub-basin. The EMDW formed in the Adriatic basin is propagating to the entire deep eastern basin so that the consequences of weak formation of this water mass in the model are observed in the whole sub-basin.

Page 9, Line 24. The greater is the mixing layer depth, the weaker is the amplitude and the peak is delayed. Is this sentence grammatically correct?

We thank the referee for this suggestion, this sentence has been changed for the sake of clarity.
**[See changes p 9, line 28-29 in the revised manuscript.]**

Page 9 and 10: To me, Fig. 7 depicts too much disagreement between simulated distributions and in situ data in many regions, not only in deep convection areas. For instance, even though it is hard to see the symbols in Fig. 7a, it seems that in surface waters of the Strait of Gibraltar the model overestimates the radiocarbon concentration by more than 20 ‰. Plus, in the discussion section it is stated that there is no time series data of 14C concentration in that area, while in the graph there are at least 4 measurements in the gulf of Cadiz and within the channel of the Strait. Could have they been used to fuel the model? In addition, explanation of data indicated in Figs 7 and 8 is confused, as description of patterns jumps from one to another without a logic sequence.

Although we partly agree with the reviewer, we think that the disagreement between simulated distribution and in situ data is particularly evident in deep convection areas as represented in Fig. 7.

In this simulation we used the same parametrization for the whole basin with same boundary conditions at the surface (first level), with $^{14}$C and the atmospheric $CO_2$ values extracted from Orr et al. (2001). The radiocarbon values in the buffer zone are prescribed from a global

simulation of radiocarbon by Mouchet et al (2016), and the ocean $^{14}C$ is initially set to a constant value of 0.85 ($\Delta^{14}C$ = -150 ‰, appropriate for the deep ocean; (Key et al., 2004)).

Hence we did not use any in-situ measurements to fuel the model for a specific region, because here we aimed to develop and optimize a $^{14}C$ modelling method for the whole Mediterranean Sea basin. Moreover no time series exists close to Gibraltar exist to force the model in the duration of the simulation. Data in the gulf of Cadiz represent a single date. As we used in-situ data to evaluate our results they cannot be employed to force the simulation.

Figures 7 and 8 present the same in-situ data of METEOR M84/3. Fig.7 provides a descriptive overview of the global horizontal distribution of $^{14}C$, where the vertical profile in Fig.8 permits to quantify the difference between the model and in-situ data at different levels. Hence the description of these figures was mad at the same time in the text.

Page 10: The radiocarbon time evolution spans from 1925 to 2008, why this particular year? In fact, Fig. 7 shows comparison between the model outputs and data of the 2011 Meteor cruise, which included measurements throughout the whole basin. Why the simulated evolution does not run until then? It would be interesting to confirm that evolution follows the pattern indicated in Fig. 7. Also, I would keep the same vertical scale in all plots to facilitate comparisons. The response found in intermediate-deep waters of the gulf of Lions is somehow unexpected, as deep convection events during winter should favor the sink of radiocarbon, particularly in extreme winters, such as that occurring in the area in 2004/2005. In Fig 9d, the intermediate layer of the gulf of Lions exhibit the lowest radiocarbon levels after the bomb episodes and deeper waters are characterized by values even lower than those found in the Tyrrhenian sub-basin. Is there any explanation for that? Moreover, are data in plot 9d integrated values through the whole water column? These results are not explained in the text. Plus, the title is wrong, it should say whole water columns.

The model was integrated from 1765 to 2011 as mentioned in section 2.3 page 7 line 14. The $^{14}C$ evolution was plotted from 1925 to 2008 in Fig.9 just to zoom on the period affected by the Suess effect (SUESS, 1955), and the entire radiocarbon transient generated by the atmospheric nuclear weapon tests performed in the 1950s and early 1960s as well as the anthropogenic $CO_2$ increase.

The intermediate-deep waters of the Gulf of Lions, characterized by $^{14}C$ values that are lower compared to the other sub-basins, are the result of a mixture of local water masses with Levantine Intermediate Water that is formed in the Levantine sub-basin, i.e. this water mass is isolated from the bomb signal in the atmosphere until arrived to the Gulf of Lion (deep convection area), hence the peak-bomb appears delayed in this area.

The data in plot 9d integrate values through the whole water column and present the same pattern as in Fig.9c for the deep water where the content of radiocarbon in the deep water control the distribution of $\Delta^{14}C$ in the whole water column.

This has been clarified in the revised text, and the title has been corrected as suggested by the reviewer. However we didn't use the same scale, because in the vertical section $\Delta^{14}C$ value up to -70 (due to the AdDW waters shortcomings, as mentioned provisory), and if we use the same

scale for the horizontal maps, many information will be not clear as well (i.e. the gradient between the different basins).

**[See changes p 11, line 3-4 in the revised manuscript.]**

Page 11, Line 10. The radiocarbon simulations provide independent and additional constraints on the thermohaline circulation and deep-water ventilation in the Mediterranean Sea. I do not see this in the manuscript. It would be the other way around. Data are interpreted according to the general circulation mechanisms known to proceed in the Med Sea.

Unlike the other tracers (e.g. CFC and Tritium), radiocarbon simulation provide additional constraints on the thermohaline circulation from the seasonal cycle to decadal and centennial timescales (e.g. Naegler, 2009; Muller et al., 2006; Rodgers et al., 1997; Guilderson et al., 1998).

In this study we have implemented the $^{14}$C module in high resolution regional model, and we work mainly on the validation on this $^{14}$C modelling method in the Mediterranean Sea basin, this work will allow many other applications especially in paleo-context e.g. Sapropel events. However direct comparison with in-situ $^{14}$C data is a new constraint for the model and it has revealed or confirmed some shortcomings such the weak EMDW formation. This will be clarified in the conclusion of the paper.

Clarified in the revised manuscript
**[See changes p 3, line 2-7 in the revised manuscript.]**

Page 12. The comparison between the model outputs and the 14C values from insitu data reported by Broecker and Gerard (1969), Stuiver et al. (1983) and Tanhua et al. (2013) reveals a good model performance in simulating the bomb/post-bomb radiocarbon distribution (Fig. 4b, Fig. 8). However the representation of the pre-bomb distribution is more contrasted in the simulation (Fig. 4a). I do not understand this paragraph. In fact, those two figures in particular show the largest disagreements, particularly in intermediate-deep waters and for the bomb-produced radiocarbon.

The reviewer is correct, there is a quite large disagreement between the model and in-situ data in some regions at intermediate-deep water depths for the bomb produced radiocarbon. However the mentioned sentence refers to Fig.6 where an important disagreement (~ 15 ‰) is observed between the model outputs and the sea-surface $^{14}$C record obtained from a 50-year-old shallow-water coral from Tisnérat-Laborde et al., (2013) for the natural pre-bomb signal. On the other hand the model niceally represents the bomb signal with the good timing and the amplitude of the peak in the near-surface water compared to in-situ data (Fig.6).

For the sake of clarity, this sentence has been modified in the revised version.
**[See changes p 12, line 6-9 in the revised manuscript.]**

Page 13, Line 7: with higher convection occurring especially during the bomb peak. Where is this shown in the paper?

We agree with the reviewer that the sentence is not clear. There is no higher convection during the period of bomb peak, but the surface water masses undergo transfer or convection with different intensity in the different sectors of the Mediterranean Basin. This argument has been clarified in the manuscript

**[See changes p 13, line 16-17 in the revised manuscript.]**

Conclusions

The natural distribution of 14C in the Mediterranean Sea is mainly affected by the inflow of Atlantic water through the Strait of Gibraltar.

As far as I understood, the concentration of radiocarbon in the Atlantic inflow did not come from in situ data or available measurements since it was taken from previous modeling approaches (as indicated in different sections of the paper). Therefore, this study does not show per se, the influence of the Atlantic radiocarbon on the distribution of this tracer in the Med Sea, as it is a fixed value used to fuel the model. The paper actually demonstrates that the entry of Atlantic waters is essential for water masses formation and circulation in the Mediterranean, which is a very well-known topic and which, in turn, regulates the distribution of radiocarbon. In fact, it would have been interesting to perform the same simulations by changing for instance the values of the water masses transport through the Strait or the radiocarbon concentration associated to the Atlantic jet. To me, such conclusion cannot be drawn from the data. I would omit it here and in the abstract or at least, the sentence should be re-written.

Unfortunately, there is no time series data of $^{14}$C concentration close to the Strait of Gibraltar. Hence simulated $^{14}$C levels in the model's Atlantic water (AW) are determined from global model estimates. As mentioned in the paper, we have performed sensitivity tests on the imposed value (as presented in Fig.1, below).
We have performed two simulations with different boundary conditions at Gibraltar (red and blue boxes and lines); the first time series (red box) gives very low level of radiocarbon in the Mediterranean Sea (as represented in Fig.2, below). In the second simulation we used a larger box (blue in Fig.1), where results are more realistic compared to some data from the North Atlantic (Tisnérat-Laborde et al., 2013; Tisnerat-Laborde, personal communication). The radiocarbon simulation greatly improves when using the larger box as boundary conditions, hence this was used to simulate $^{14}$C in the Mediterranean Sea.

This point has been also raised by reviewer #1 and it has been addressed in the revised version of the manuscript.

**[See changes p 7, line 18-25 in the revised manuscript.]**

[Figure]

Fig. 1: The concentration of radiocarbon in the Atlantic inflow (NEMO global model, Mouchet et al. 2016).

[Figure]

Fig.2: $\Delta^{14}$C values (in ‰) in the Ligurian sub-basin from 1765 to 2008 for the surface water (0-10 m depth), together with available in-situ observations (Tisnérat-Laborde et al., 2013) from coral (black dashed line). Simulated data obtained using the smaller box (blue dashed line, simu 2), and the larger box (blue line, simu 1).